# Combining RAS(ON) G12C-selective inhibitor with SHP2 inhibition sensitises lung tumours to immune checkpoint blockade

Panayiotis Anastasiou[1], Christopher Moore [1], Sareena Rana[1], Mona Tomaschko[1], Claire E. Pillsbury[1], Andrea de Castro[1], Jesse Boumelha[1], Edurne Mugarza[1], Sophie de Carné Trécesson [1], Ania Mikolajczak[2], Cristina Blaj[3], Robert Goldstone[4], Jacqueline A. M. Smith[3], Elsa Quintana[3], Miriam Molina-Arcas [1] ✉ & Julian Downward [1] ✉

Mutant selective drugs targeting the inactive, GDP-bound form of KRAS[G12C] have been approved for use in lung cancer, but resistance develops rapidly. Here we use an inhibitor, (RMC-4998) that targets RAS[G12C] in its active, GTP-bound form, to treat KRAS mutant lung cancer in various immune competent mouse models. RAS pathway reactivation after RMC-4998 treatment could be delayed using combined treatment with a SHP2 inhibitor, which not only impacts tumour cell RAS signalling but also remodels the tumour micro-environment to be less immunosuppressive. In an immune inflamed model, RAS and SHP2 inhibitors in combination drive durable responses by suppressing tumour relapse and inducing development of immune memory. In an immune excluded model, combined RAS and SHP2 inhibition sensitises tumours to immune checkpoint blockade, leading to efficient tumour immune rejection. These preclinical results demonstrate the potential of the combination of RAS(ON) G12C-selective inhibitors with SHP2 inhibitors to sensitize tumours to immune checkpoint blockade.

KRAS oncogenic mutations are very frequent in lung cancer and the leading causes of cancer-related deaths worldwide[1,2]. KRAS proteins act as signalling hubs where upstream receptor tyrosine kinase (RTK) activation enables guanine exchange factors (GEF) to facilitate the shift of KRAS from the GDP-bound inactive state to the GTP-bound active state[3]. Glycine to cysteine substitutions at codon 12 (G12C) are the most prevalent KRAS oncogenic mutations in lung cancer[4]. In contrast to other codon 12 mutations, G12C mutations do not majorly impair the intrinsic GTPase hydrolytic activity of KRAS but do render KRAS[G12C] insensitive to canonical GTPase-activating proteins (GAP) and thus drive the cycling KRAS[G12C] to exist predominantly in its active GTP-bound form[5]. KRAS[G12C] mutant-specific inhibitors such as adagrasib (MRTX849) and sotorasib (AMG 510) have been developed which

covalently bind to the mutant cysteine residue of GDP-bound KRAS[G12C] and block it in its inactive state, preventing it from cycling back to the active GTP-bound state[6,7].

The promising initial clinical responses with KRAS[G12C] inhibitors adagrasib and sotorasib led to their approval for clinical use for advanced KRAS[G12C] mutant non-small cell lung cancer (NSCLC). However, tumours quite quickly develop resistance to these drugs with median progression-free survivals under 7 months[8–10]. Several mechanisms of acquired and adaptive resistance have been suggested which underlines the necessity for developing combination therapies to potentiate the effects of KRAS[G12C] inhibitors[11,12]. A number of preclinical studies have shown that KRAS[G12C] inhibition results in feedback reactivation of upstream RTKs which can result in the activation of

[1]Oncogene Biology Laboratory, Francis Crick Institute, London, UK. [2]Experimental Histopathology, Francis Crick Institute, London, UK. [3]Revolution Medicines, Inc., Redwood City, CA, USA. [4]Bioinformatics & Biostatistics Science Technology Platform, Francis Crick Institute, London, UK. ✉e-mail: miriam.molina@crick.ac.uk; julian.downward@crick.ac.uk

wild-type and mutant RAS isoforms. Therefore, inhibition of SHP2, which mediates upstream signalling from RTKs to RAS, is an alternative mechanism to overcome RTK-mediated adaptive resistance[13–19]. Elevated expression of mutant KRAS in its active form has also been proposed as a mechanism of resistance to KRAS[G12C] (OFF) inhibitors, thus this could be overcome by using RAS(ON) G12C-selective inhibitors[20].

Recently, RAS[G12C] inhibitors have been developed that act as non-degrading molecular glues to create an inactive tri-complex with cyclophilin A (CYPA) and the GTP-bound, active form of KRAS[G12C]. These have been termed RAS(ON) G12C-selective inhibitors and include RMC-4998 which is a preclinical tool compound representative of the investigational agent RMC-6291. The binding of these agents to the active state of KRAS may resolve some limitations of the inactive state selective KRAS[G12C](OFF) inhibitors[21]. RMC-6291 is currently undergoing clinical evaluation[22] and while duration of response results are awaited, based on experience to date with targeted cancer therapies it is likely that tumours will also develop resistance eventually, meaning that combination strategies will be needed.

The specificity of mutant-specific KRAS[G12C] inhibitors for mutant KRAS in cancer cells without inhibiting wild-type KRAS in normal cells has helped to define the role of oncogenic KRAS signalling in modulating anti-tumour immune responses. This includes suppression of interferon (IFN) signalling, cytokine expression and subsequent recruitment of immunosuppressive myeloid populations and suppression of adaptive T cell-mediated anti-tumour immune responses[23,24]. Numerous studies have demonstrated that KRAS inhibition reverses immune suppression, causes a profound remodelling of the tumour immune microenvironment (TME) and activates anti-tumour immunity, which provides a window of opportunity for combination with immunotherapies[24–26]. In contrast with targeted therapies, immunotherapies result in long-term responses in a small fraction of patients[27]. Moreover, anti-PD-1/anti-PD-L1 therapies are the first line of treatment for advanced NSCLC[28,29]. Based on that, several clinical trials are exploring combinations of KRAS[G12C] inhibitors with anti-PD-1/anti-PD-L1 immune checkpoint blockade (ICB). However, using preclinical models, we have recently demonstrated that this combination is only beneficial in immunogenic tumours, characterised by high lymphocyte infiltration and some baseline sensitivity to anti-PD-1, while no combination benefit was observed in immune evasive tumours, which lack lymphocyte infiltration and ICB sensitivity[24]. Therefore, additional combination therapies that can reshape the TME further may be required to achieve long-term responses using KRAS[G12C] inhibitors, in particular in patients with immune cold tumours. One possible therapy for such combinations might be inhibitors of SHP2: these not only suppress the adaptive rewiring of signalling within the cancer cells that diminishes sensitivity to KRAS[G12C] inhibition, but can also directly affect signalling of non-cancer cells within the TME and enhance anti-tumour immune responses[15,30–33].

In this work, we combine the active state selective RAS(ON) G12C-selective compound RMC-4998 with the SHP2 inhibitor RMC-4550 and ICB and investigate the effects of these combinations in preclinical models of lung cancer with varying degrees of immunogenicity. We demonstrate that MAPK reactivation in the cancer cells is unavoidable even when using a compound targeting KRAS[G12C] active state, although this can be suppressed and delayed with the addition of a SHP2 inhibitor. Furthermore, we show that the combination of RMC-4998 and RMC-4550 generates long-term responses in mice with immunogenic lung tumours, including a high proportion of immune-mediated tumour eradication. In mice with aggressive immune-excluded lung tumours, combination of RMC-4998 and RMC-4550 reshapes the TME, activates adaptive immunity and sensitises tumours to ICB, thus converting these non-inflamed tumours to a more inflamed, ICB-sensitive phenotype. Finally, we provide evidence that suggests direct effects of SHP2 inhibitor on cells of the TME.

## Results

### Combination of RAS[G12C](ON) and SHP2 inhibition suppresses MAPK reactivation and promotes IFN responses

In order to investigate the tumour cell intrinsic effects of the RAS(ON) G12C-selective tri-complex inhibitor, RMC-4998, we initially performed cell viability assays using KRAS[G12C]-mutant NSCLC cell lines treated with either RMC-4998 or the KRAS[G12C](OFF) inhibitor adagrasib (MRTX849). Treatment of both human (CALU1 and NCI-H23) and murine (KPAR[G12C] and 3LL-ΔNRAS) NSCLC cell lines with either compound led to a concentration-dependent decrease in cell viability, with RMC-4998 displaying increased activity compared to MRTX849 (Fig. 1a and Supplementary Fig. 1a). Next, we assessed the effect of both compounds on downstream signalling. MAPK activity was rapidly abrogated with RMC-4998, indicated by a reduction in ERK phosphorylation within the first 15 min of incubation, whereas MAPK inhibition with MRTX849 was only evident at later time points (Supplementary Fig. 1b, c). The RAS(ON) G12C-selective inhibitor thus acts more rapidly to block KRAS signalling than the KRAS[G12C](OFF) inhibitor, as expected since it is directly targeting the active form of KRAS. However, at longer time points, targeting the active state of KRAS[G12C] with RMC-4998 also showed a recovery of ERK phosphorylation at 24–48 h that was more comparable to that seen with MRTX849, indicating reactivation of the MAPK pathway and development of adaptive mechanisms (Fig. 1b). These observations suggest that even using inhibitors that target the active state of KRAS[G12C], adaptive mechanisms will eventually develop, emphasising the necessity for combinatorial strategies.

Previous studies have demonstrated the role of SHP2 in promoting MAPK pathway reactivation by propagating signalling downstream of elevated upstream RTK activity including inducing cycling of wild-type K-, H- and N-RAS isoforms to the GTP-bound activated state[14]. For this reason, we combined RMC-4550, a SHP2 allosteric inhibitor[32], with the RAS(ON) G12C-selective inhibitor RMC-4998 to assess if MAPK reactivation could be suppressed. Indeed, in human NSCLC cell lines combination of RMC-4550 with RMC-4998 reduced the rebound of ERK phosphorylation at later time points (24–48 h) and resulted in a stronger reduction in cell viability (Fig. 1c, d). Moreover, this combination led to enhanced pathway inhibition, shown by decreased expression of downstream MAPK transcript target *Dusp6*, decreased cell viability and increased apoptosis in the mouse cell line KPAR[G12C] (Fig. 1e, f and Supplementary Fig. 1d). These effects extended to longer time points (6 days), where KPAR[G12C] cultures were replenished with compounds every 48 h, validating previous observations that pathway reactivation is not due to reduced compound availability (Fig. 1g).

KRAS inhibition can also lead to increased activation of type I and II IFN responses in cancer cells, which are crucial for anti-tumour immune responses[24]. Consistent with this, treatment with RMC-4998 enhanced IFNγ transcriptional responses measured by the increased expression of IFN target genes (*Irf1*, *Irf7*, *Irf9*, *Cd274*, *B2m*, *H2-d1*) in response to in vitro IFNγ treatment (Fig. 1h and Supplementary Fig. 1e). Interestingly, SHP2 inhibition, using RMC-4550, led to similar effects although to a lesser extent, due to a less efficient suppression of the MAPK pathway (Fig. 1c) while combination phenocopied RMC-4998 (Fig. 1h and Supplementary Fig. 1e).

In summary, these data demonstrate that in KRAS[G12C] mutant lung cancer cells, targeting together RAS[G12C](ON) and SHP2 limits the development of adaptive resistance, which results in increased cancer cell apoptosis and tumour cell intrinsic IFN pathway responses.

### RAS[G12C](ON) and SHP2 inhibition drive immune-dependent cures in an immunogenic model of NSCLC

Given the crucial role of KRAS in modulating anti-tumour immune responses, we determined the anti-tumour activity of RMC-4998 and RMC-4550 in treating tumours formed by transplantation of KPAR[G12C]

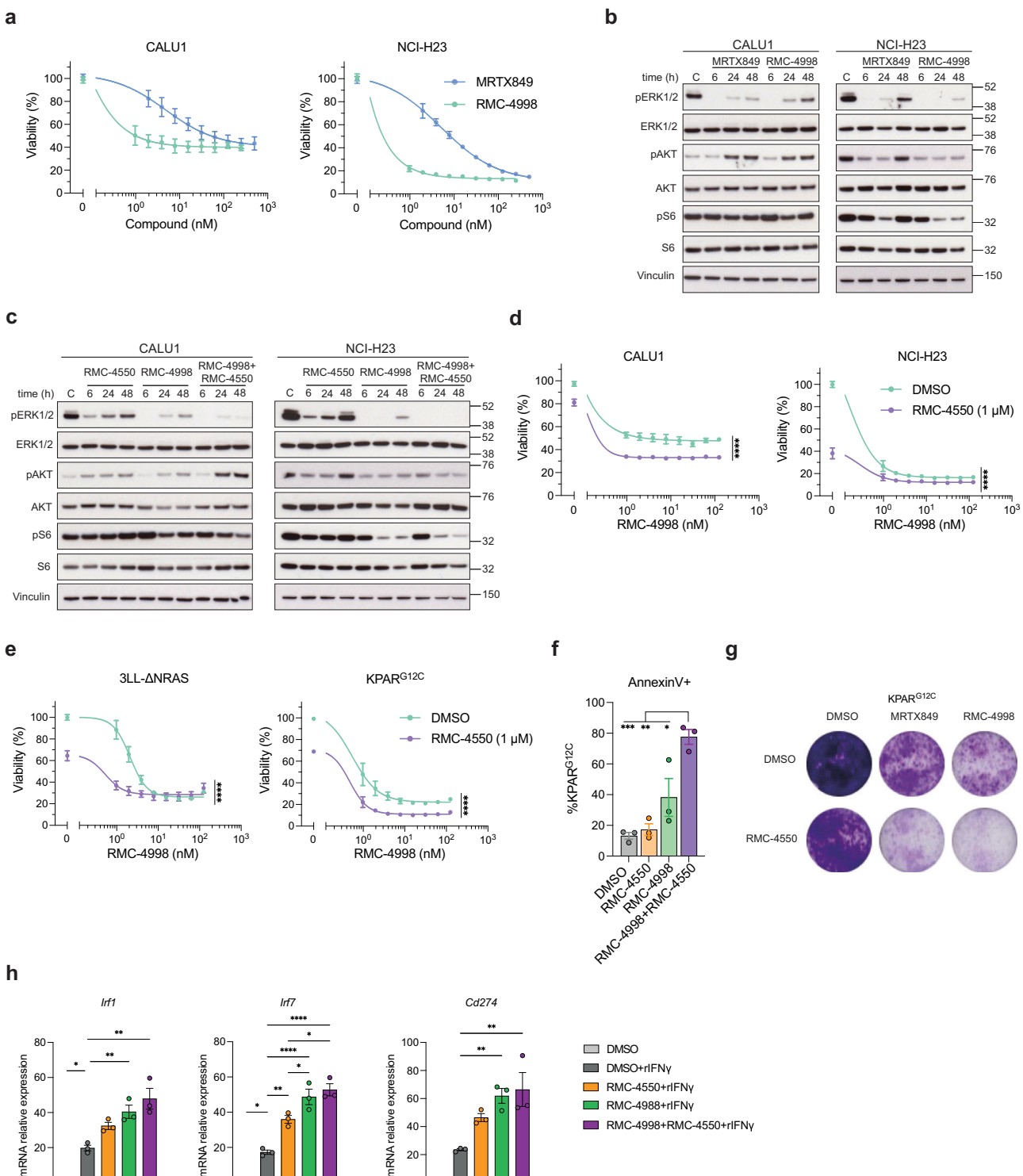

**Fig. 1 | SHP2 inhibitor RMC-4550 prevents adaptive response to the RAS(ON) G12C-selective inhibitor RMC-4998. a** Viability of human KRAS-mutant cell lines treated with serial dilutions of RMC-4998 or MRTX849 for 72 h. Data are mean ± SEM of two independent experiments. **b** Western blot of human KRAS-mutant cell lines treated for 6, 24 or 48 h with 100 nM MRTX849 or 100 nM RMC-4998. DMSO-treated cells were used as control (C). Blot is representative of two independent experiments. **c** Western blot of human KRAS-mutant cell lines treated at different time points with 1 μM RMC-4550, 100 nM RMC-4998 or the combination. Blot is representative of three independent experiments. Viability of human (**d**) or mouse (**e**) KRAS-mutant cell lines treated for 72 h with serial dilutions of RMC-4998 in the presence or absence of 1 μM RMC-4550. Data are mean ± SEM of three independent experiments. Statistics were calculated using two-way ANOVA. **f** Percentage of

annexin V positive KPAR[G12C] cells after 72-h treatment with 1 μM RMC-4550, 100 nM RMC-4998 or the combination. Data are mean ± SD of three independent experiments. Analysis was done using one-way ANOVA. **g** Crystal violet staining of KPAR[G12C] cells treated for 6 days with 200 nM MRTX849, 200 nM RMC-4998, 1 μM RMC-4550 or the combination. Representative image of two independent experiments. **h** qPCR analysis of IFN-induced genes in KPAR[G12C] cells treated for 24 h with 1 μM RMC-4550, 100 nM RMC-4998 or the combination, in the presence of 100 ng/ml recombinant IFNγ. DMSO-treated cells are used as control. Data are mean ± SD of three independent experiments. Analysis was done using one-way ANOVA. Only significant comparisons are shown. The DMSO condition has only been compared to DMSO + rIFNγ. For all statistical analysis $*p < 0.05$, $**p < 0.01$, $***p < 0.001$, $****p < 0.0001$. Source data and exact $p$ values are provided as a Source Data file.

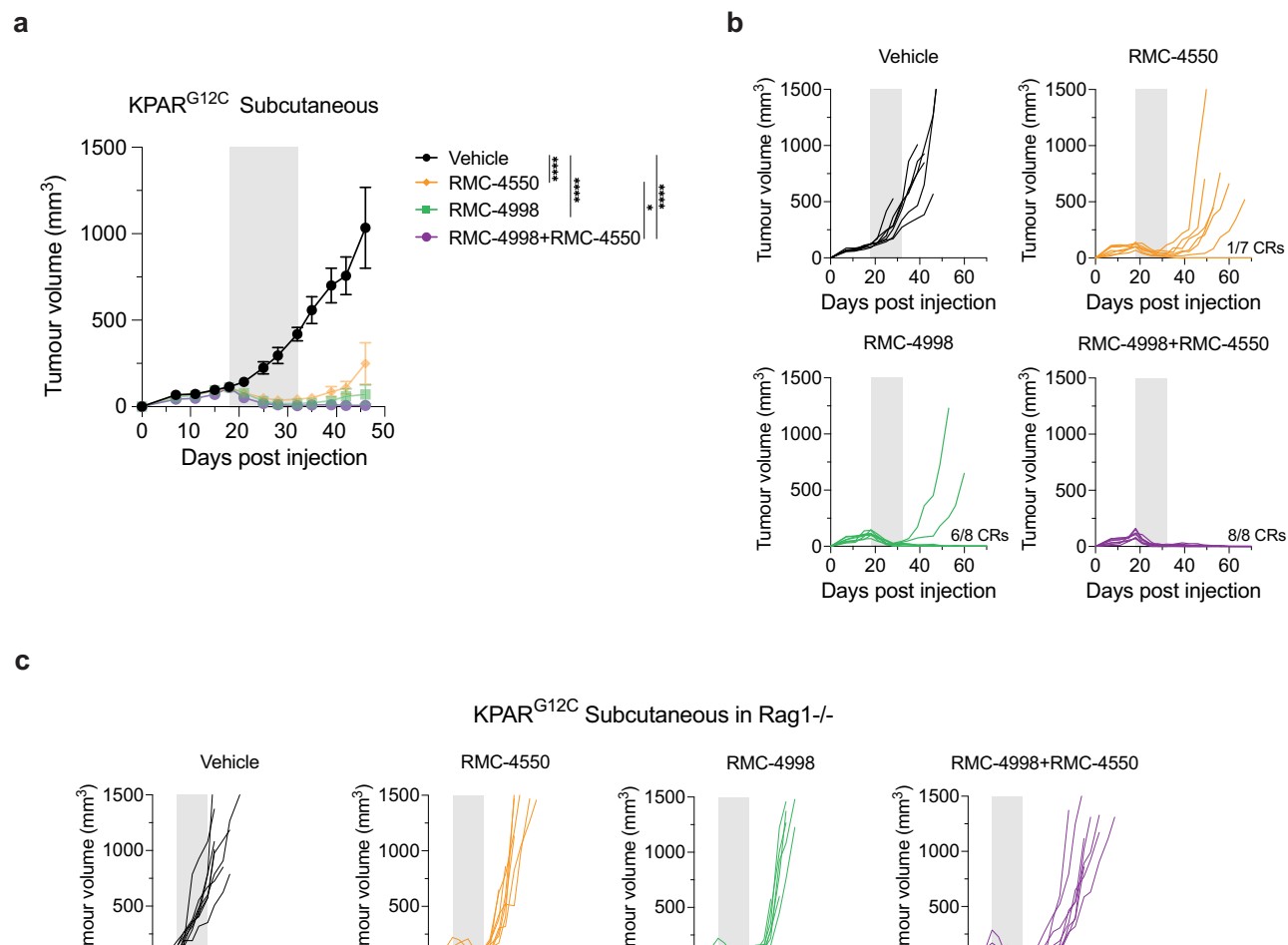

**Fig. 2 | Combination of the RAS$^{G12C}$(ON) inhibitor RMC-4998 with the SHP2 inhibitor RMC-4550 in immunogenic subcutaneous KPAR$^{G12C}$ tumours.**
**a** Tumour growth of KPAR$^{G12C}$ subcutaneous tumours treated daily for 2 weeks with 30 mg/kg RMC-4550 and/or 100 mg/kg RMC-4998. Vehicle ($n = 7$), RMC-4550 ($n = 7$), RMC-4998 ($n = 8$), RMC-4998 + RMC-4550 ($n = 8$). Grey area indicates treatment period. Data are mean tumour volumes ± SEM. Analysis was performed using one-way ANOVA (*$p < 0.05$, ****$p < 0.0001$). **b** Individual tumour volumes for

the full duration of the experiment of mice in (**a**). Number of complete regressions (CR) is indicated. Grey area indicates treatment period. **c** Individual tumour growth of KPAR$^{G12C}$ subcutaneous tumours grown in Rag1$^{-/-}$ mice treated daily for 2 weeks with 30 mg/kg RMC-4550 and/or 100 mg/kg RMC-4998. Vehicle ($n = 8$), RMC-4550 ($n = 8$), RMC-4998 ($n = 7$), RMC-4998 + RMC-4550 ($n = 8$). The average of tumour growth and statistical significance is shown in Supplementary Fig. 2b. Source data and exact $p$ values are provided as a Source Data file.

cells, an immunogenic mouse model of NSCLC[26]. These cells, derived from a KRAS$^{LSL-G12D/-}$Trp53$^{f/fl}$ Rosa26$^{hA3Bi}$ Rag1$^{-/-}$ background, stimulate endogenous anti-tumour immune responses directed against derepressed endogenous retroviral antigens and are partially responsive to immunotherapy[26]. Notable tumour shrinkage and generation of some durable complete regressions (CRs, where no tumour reoccurrence occurs even after treatment withdrawal) of established subcutaneous KPAR$^{G12C}$ tumours in immunocompetent mice were observed with both RMC-4998 and RMC-4550 (Fig. 2a, b). Specifically, after only 2 weeks of treatment, RMC-4998 administration led to generation of 6/8 (75%) CRs whilst RMC-4550 led to generation of 1/7 (14.3%) CRs. Importantly, combination of RMC-4998 and RMC-4550 prevented tumour relapse after treatment withdrawal in all the treated mice, giving 8/8 CRs, confirming the increased activity of the combination (Fig. 2b). Mice that had shown complete tumour regression were then rechallenged on the opposite flank with KPAR$^{G12C}$ subcutaneous injections in the absence of further treatment, to assess development of immune memory against the KPAR$^{G12C}$ cancer cells. Indeed, most mice rejected tumour rechallenge and remained tumour free (Supplementary Fig. 2a). These data indicate that both RMC-4998 and RMC-

4550 promote the induction of anti-tumour immune responses. Therefore, we assessed the effects of these compounds in immunodeficient subcutaneous KPAR$^{G12C}$ tumour-bearing Rag1$^{-/-}$ mice which lack functional T and B lymphocytes and are incapable of inducing adaptive immune responses. Combination of RMC-4998 and RMC-4550 displayed a higher activity than either RMC-4550 or RMC-4998 alone. However, no generation of CRs was observed and all tumours regrew on treatment withdrawal, suggesting that adaptive immunity is essential for generation of long-term CRs (Fig. 2c and Supplementary Fig. 2b).

Recent studies have highlighted the contrast of adaptive anti-tumour immune responses between subcutaneous and orthotopic lung tumours[34]. Accordingly, we wanted to extend our findings to the orthotopic setting and investigate the effects of RMC-4998 and RMC-4550 in orthotopic KPAR$^{G12C}$ lung tumours, which have been shown to be partially sensitive to PD-1 blockade[26]. Similarly to subcutaneous tumours, inhibition of the active form of KRAS$^{G12C}$ achieved better responses than SHP2 inhibition. Micro-CT scanning at the end of the 2 weeks of treatment revealed that KRAS inhibition with RMC-4998 resulted in 100% regressions in most of the tumours (Fig. 3a).

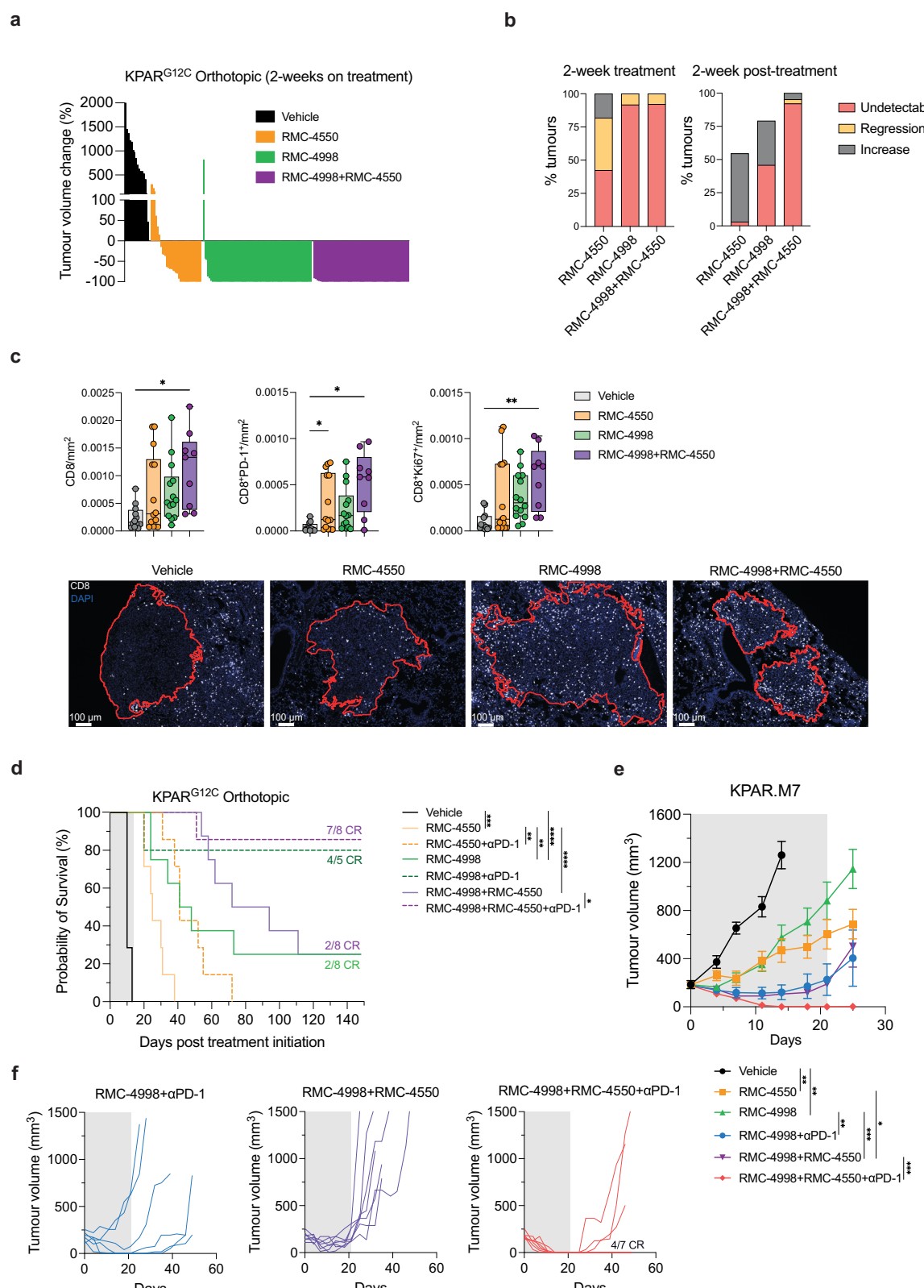

Nonetheless, micro-CT scans 2 weeks after treatment withdrawal revealed that ~50% of RMC-4998 treated tumours relapsed whereas only ~10% of RMC-4998 and RMC-4550 combination-treated tumours relapsed, demonstrating the importance of this combination in suppressing early tumour relapse (Fig. 3b). Treatment of KPAR^G12C tumours with RMC-4998 and RMC-4550 for 2 days resulted in decreased expression of *Dusp6* which indicated suppression of MAPK pathway

(Supplementary Fig. 3a), consistent with our in vitro observations (Supplementary Fig. 1d). In parallel, elevated expression of *Cd8a*, *Pdcd1* and *Gzma* and decreased expression of *Arg1* after only 2 days of treatment suggested infiltration and activation of CD8⁺ T cells and suppression of M2-like myeloid cells, respectively (Supplementary Fig. 3a). Treatment of KPAR^G12C tumours with RMC-4998 and RMC-4550 for 4 days also revealed increased numbers of proliferating and

**Fig. 3 | Combination of the RAS(ON) G12C-selective inhibitor RMC-4998 with the SHP2 inhibitor RMC-4550 and anti-PD-1 in immunogenic KPAR^G12C tumours. a** Tumour volume change of orthotopic KPAR^G12C tumours after 2 weeks of daily treatment of mice with vehicle (*n* = 2), 30 mg/kg RMC-4550 (*n* = 7), 100 mg/kg RMC-4998 (*n* = 8) or RMC-4998 + RMC-4550 (*n* = 8). Each bar represents one tumour. Same treatment doses were administered in the other panels. **b** Left panel: percentage of tumours from (**a**) after 2 weeks of treatment that were undetectable by micro-CT scan, regressing or increasing compared with the initial volume. Right panel: same tumours were classified based on volume change 2 weeks after treatment withdrawn compared with the tumour volume at the end of the treatment. Missing tumours correspond to mice that died before the 4-week scan. **c** Top: density of CD8^+ T cells, Ki-67^+ and PD-1^+ CD8 T cells identified using 8-plex immunofluorescence of orthotopic KPAR^G12C tumours after 4 days of treatment with vehicle (*n* = 5), RMC-4550 (*n* = 5), RMC-4998 (*n* = 6), RMC-4998 + RMC-4550 (*n* = 4). Data are median with box extending from the 25th to the 75th percentile

and whiskers showing min and max values. Each point is an individual tumour. Analysis was performed using one-way ANOVA. Bottom: representative images of CD8^+ T cells. **d** Survival of mice bearing KPAR^G12C orthotopic lung tumours treated daily for 2 weeks with RMC-4550 and/or RMC-4998 in the presence or absence of anti-PD-1 (4 doses in 2 weeks). Grey area indicates treatment period. Number of complete responders (CR) is indicated. *n* = 5–8 mice/group. Analysis was done using log-rank Mantel−Cox test. **e** Tumour growth of KPAR.M7 subcutaneous tumours treated with vehicle (*n* = 7), RMC-4550 (*n* = 7), RMC-4998 (*n* = 7), RMC-4998 + anti-PD-1 (*n* = 6), RMC-4998 + RMC-4550 (*n* = 7), RMC-4998 + RMC-4550+ anti-PD-1 (*n* = 7). Grey area indicates treatment period. Data are mean tumour volumes ± SEM. Analysis was performed using two-way ANOVA. **f** Individual tumour volumes for the full duration of the experiment of mice in (**e**). Number of complete regressions (CR) is indicated. Single treatment and controls are shown in Supplementary Fig. 3b. For all statistical analysis *p < 0.05, **p < 0.01, ***p < 0.001, ****p < 0.0001. Source data and exact *p* values are provided as a Source Data file.

activated CD8^+ T cells (Fig. 3c). Based on these observations and the importance of adaptive immunity for generating long-term responses, we decided to combine RMC-4998 and RMC-4550 with anti-PD-1 ICB therapy, an established immunotherapy option for NSCLC patients. Notably, addition of anti-PD-1 therapy to RMC-4550 or/and RMC-4998 extended the survival of mice with orthotopic KPAR^G12C tumours and resulted in CRs in those groups treated with the RAS(ON) G12C-selective inhibitor (Fig. 3d), even though the period of treatment was only 2 weeks. Combination of RMC-4998 with anti-PD-1 was the major driver of CRs whilst combination of RMC-4550 and RMC-4998, even though it extended the survival, only generated an equal number of complete responders to RMC-4998 monotherapy. To test if the lack of differences between the triple combination and RMC-4998 with anti-PD-1 was due to the high sensitivity of KPAR^G12C tumours to RMC-4998 (Fig. 3a), we generated a less responsive cell line, KPAR.M7, derived from an orthotopic lung KPAR^G12C tumour growing on MRTX849 (see section "Methods"). In subcutaneous KPAR.M7 tumours treatment with RMC-4998 or RMC-4550 only delayed tumour growth (Fig. 3e and Supplementary Fig. 3b) however, failed to generate CRs contrarily to what we observed in the parental KPAR^G12C model (Fig. 2b). Although RMC-4998 sensitised KPAR.M7 to anti-PD-1 (Fig. 3e), it still failed to generate any durable CRs whereas triple combination generated 4/7 (57.1%) CRs (Fig. 3f).

In summary, these data demonstrate that combination of RAS^G12C(ON) and SHP2 inhibition can activate adaptive immune responses, suppress tumour relapse, and synergise with anti-PD-1 immunotherapy to generate complete cures at high frequency in immunogenic mouse models of NSCLC.

## RAS^G12C(ON) and SHP2 inhibition synergise with ICB to generate cures in a subcutaneous immune-excluded anti-PD-1 resistant model of NSCLC

We next utilised the 3LL-ΔNRAS transplantable mouse model to evaluate responses to RMC-4998 and RMC-4550 and the potential for combination with immunotherapies in an immune-excluded, anti-PD-1 resistant lung cancer model. Initially, we assessed the effect of anti-PD-1 and/or anti-CTLA-4 on subcutaneous 3LL-ΔNRAS tumours in immunocompetent mice. Indeed, 3LL-ΔNRAS tumours were completely refractory to anti-PD-1 and/or anti-CTLA-4, with the exception of one responder in the anti-CTLA-4 group (Fig. 4a and Supplementary Fig. 4a). Treatment with either RMC-4998 or RMC-4550 resulted in a profound inhibition of tumour growth, which was extended when both compounds were combined (Fig. 4b). In contrast with the results obtained in the KPAR^G12C model (Fig. 2a, b), similar responses were generated using either RMC-4998 or RMC-4550. Moreover, treatment with these targeted therapies did not lead to any durable CRs as all mice relapsed while on treatment, including those treated with RMC-4998 and RMC-4550 together (Fig. 4b, c). Of note, addition of anti-PD-1

to the combination of RMC-4998 and RMC-4550 enhanced anti-tumour responses as it led to generation of CRs in 3/8 (37.5%) of mice. In contrast, combination of anti-PD-1 with either RMC-4998 or RMC-4550 alone showed only marginal enhancement of anti-tumour responses (Supplementary Fig. 4b). These data reinforce the previous observations that KRAS^G12C inhibition alone does not sensitise immune-excluded tumours to anti-PD-1 and underlines the prerequisite for inhibition of both KRAS^G12C and SHP2 inhibition for sensitising 3LL-ΔNRAS tumours to anti-PD-1 immunotherapy[24]. In contrast, both RMC-4998 or RMC-4550 displayed increased anti-tumour activity and induced occasional CRs when combined as single agents with anti-CTLA-4 (Supplementary Fig. 4b). However, addition of anti-PD-1 did not much enhance the activity of these doublets (Supplementary Fig. 4b). The additional benefit observed with anti-CTLA-4 in subcutaneous tumours may reflect depletion of T regulatory cells (Tregs) or attenuation of CTLA-4-mediated inhibition of positive co-stimulation of effector T cells by CD28[35]. Interestingly, the combination of anti-CTLA-4 with RMC-4998 and RMC-4550 displayed a similar effect to the triple combination with anti-PD-1 (Supplementary Fig. 4d), whereas the quadruple combination of targeted and ICB therapies led to tumour eradication in almost all mice (Fig. 4c, e and Supplementary Fig. 4d). The complete responders generated were rechallenged on the opposite flank with 3LL-ΔNRAS subcutaneous injection, without further therapeutic intervention. Importantly, we observed tumour rejection in most mice, which indicated the development of effective immune memory in this immune evasive cancer model (Fig. 4d and Supplementary Fig. 4c).

We then sought to analyse if the combination of the KRAS^G12C(OFF) inhibitor adagrasib (MRTX849) with SHP2 inhibition could also sensitise immune-excluded tumours to anti-PD-1. Similarly to the results obtained with the RAS(ON) G12C-selective inhibitor RMC-4998, RMC-4550 extended the survival of mice treated with MRTX849 (Supplementary Fig. 4e) and the addition of anti-PD-1 to this combination resulted in the generation of durable CRs (Supplementary Fig. 4e). To extend our findings, we made use of the subcutaneous KPB6^G12C model which develops immune cold tumours, due to low number of clonal single-nucleotide variants, that do not respond to combination of KRAS^G12C inhibition and anti-PD-1 treatment even though they are highly sensitive to KRAS^G12C inhibition[24,26]. Similarly to what previously observed, RMC-4998 treatment of subcutaneous KPB6^G12C tumours induced significant tumour growth inhibition (Fig. 4f). Treatment with RMC-4998 and RMC-4550 did not provide added benefit, whereas addition of anti-PD-1 to the RMC-4998 and RMC-4550 combination suppressed tumour growth further (Fig. 4f), indicating that this combination partially sensitises this immune cold model to anti-PD-1.

In summary, in the subcutaneous setting, combined RAS^G12C(ON) and SHP2 inhibition renders immune-excluded NSCLC tumours

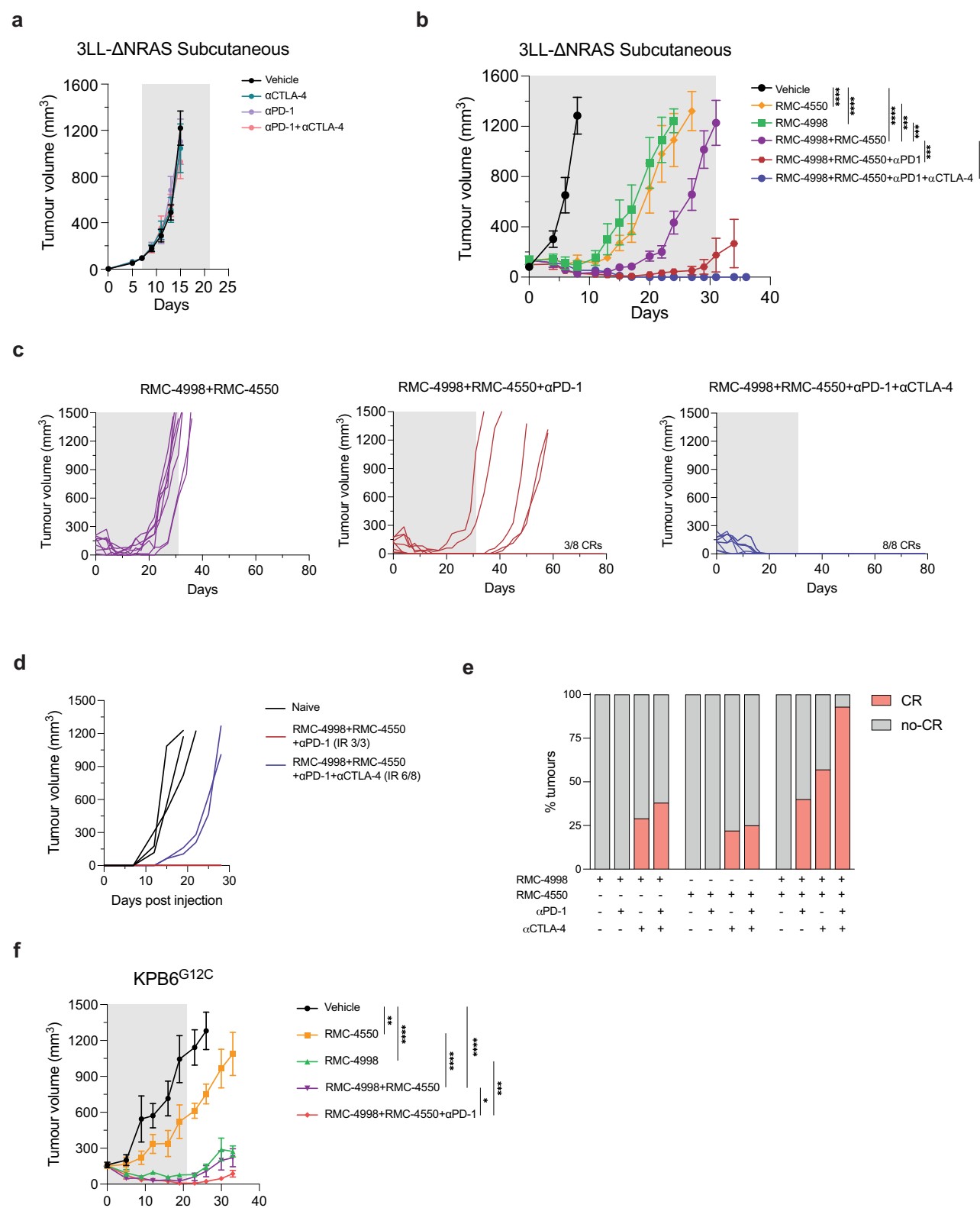

responsive to anti-PD-1 immunotherapy, with further addition of anti-CTLA-4 causing eradication of most tumours.

## RAS^G12C(ON) and SHP2 inhibition reshape an immune-excluded lung cancer TME towards an inflamed phenotype

Previous studies have shown that both KRAS^G12C(OFF) and SHP2 inhibitors can reshape the immune TME and partially reverse immune

suppression[15,30]. Therefore, we used the orthotopic 3LL-ΔNRAS model to investigate the effects of the RAS(ON) G12C-selective inhibitor RMC-4998 and the SHP2 inhibitor RMC-4550 on the lung TME. 3LL-ΔNRAS tumours have been characterised by a predominant infiltration of myeloid cells that suppress immune responses and promote tumour growth. Flow cytometric analysis of established 3LL-ΔNRAS lung tumours validated this along with poor infiltration of lymphocytes in

**Fig. 4 | Combination of RAS(ON) G12C-selective inhibitor RMC-4998 with SHP2 inhibitor RMC-4550 sensitises 3LL-ΔNRAS subcutaneous tumours to immunotherapies. a** Tumour growth of 3LL-ΔNRAS subcutaneous tumours treated with 10 mg/kg anti-PD-1, 5 mg/kg anti-CTLA-4 or the combination. Vehicle ($n = 6$), anti-CTLA-4 ($n = 8$), anti-PD-1 ($n = 7$), anti-PD-1 + anti-CTLA-4 ($n = 6$). Antibodies were administered twice a week for 2 weeks. Grey area indicates treatment period. Same ICB doses and treatment schedules were administered in the other panels. Data are mean tumour volumes ± SEM. Analysis was performed using two-way ANOVA. Only significant comparisons are shown. **b** Tumour growth of 3LL-ΔNRAS subcutaneous tumours treated daily with 30 mg/kg RMC-4550 and/or 100 mg/kg RMC-4998 in presence or absence of ICB. Vehicle ($n = 7$), RMC-4550 ($n = 6$), RMC-4998 ($n = 8$), RMC-4998 + RMC-4550 ($n = 8$), RMC-4998 + RMC-4550 + anti-PD-1 ($n = 8$), RMC-4998 + RMC-4550 + anti-PD-1 + anti-CTLA-4 ($n = 8$). Grey area indicates treatment period. Data are mean tumour volumes ± SEM. Analysis was performed using two-way ANOVA test. **c** Individual tumour volumes for the full duration of the experiment of mice in (**b**). Number of complete regressions (CR) is indicated. **d** Mice in (**c**) that rejected the primary tumour were rechallenged on the opposite flank and tumour volume was measured. Number of mice that achieved immune rejections (IR) is indicated. Naïve mice of similar age were used as control. Legends indicate the treatment that the primary tumour received. **e** Graph showing the percentage of complete regressions (CR) of mice in (**c**) and Supplementary Fig. 4b, d. **f** Tumour growth of KPB6^G12C subcutaneous tumours treated daily for 3 weeks with 30 mg/kg RMC-4550 and/or 100 mg/kg RMC-4998 in presence or absence of 10 mg/kg anti-PD-1. Vehicle ($n = 6$), RMC-4550 ($n = 7$), RMC-4998 ($n = 7$), RMC-4998 + RMC-4550 ($n = 7$), RMC-4998 + RMC-4550 + anti-PD-1 ($n = 8$). Grey area indicates treatment period. Data are mean tumour volumes ± SEM. Analysis was performed using two-way ANOVA. For all statistical analysis *$p < 0.05$, **$p < 0.01$, ***$p < 0.001$, ****$p < 0.0001$. Source data and exact $p$ values are provided as a Source Data file.

contrast to the immunogenic KPAR^G12C model which was predominantly infiltrated by lymphocytes (Supplementary Fig. 5a). Next, we sought to immunophenotype 3LL-ΔNRAS tumours after 8 days of treatment with RMC-4998 and/or RMC-4550 with a focus on myeloid cells. Treatment revealed reduced infiltration of myeloid cells, such as classical monocytes (Classical MOs; CD11b^+CD11c^−Ly6G^−Ly6C^+) and neutrophils (Neutrophils1; CD11b^+CD24^+Ly6G^high Ly6C^+; Fig. 5a). Different effects between compounds were observed in the interstitial macrophages (MΦs; CD64^+MerTK^+), where RMC-4998 increased the infiltration while this population was reduced with the combination (Fig. 5a). RMC-4998 treatment also increased the presence of CD86^+MHCII^+ and PD-L1^+ interstitial MΦs with the RMC-4550 treated tumours having a distinct increased proportion of these subsets (Fig. 5b). These observations hinted towards some RMC-4550 effects on the myeloid compartment of the TME independent of the cancer-cell intrinsic MAPK pathway. Further characterisation of the interstitial MΦ compartment revealed increased expression of MHCII, PD-L1, CD11c and MerTK and suppressed expression of CD206 in the RMC-4550 treated groups, suggesting a functional conversion of interstitial MΦs towards maturation and antigen presentation, to be mainly driven by SHP2 inhibition (Fig. 5c and Supplementary Fig. 5b). In contrast, interstitial MΦs of tumours treated with either RMC-4998 or RMC-4550 had reduced ARG1 expression, suggesting that cancer-cell MAPK pathway inhibition hindered immunosuppressive properties of interstitial MΦs (Fig. 5c, d and Supplementary Fig. 5b). Interestingly, only the combination of RMC-4998 and RMC-4550 induced elevated expression of *Nos2*, indicating that although either KRAS^G12C or SHP2 inhibition can drive depletion of immunosuppressive myeloid-driven programmes, only combined inhibition results in the induction of direct anti-tumour myeloid functions (Fig. 5d). RNA sequencing (RNA-seq) of treated 3LL-ΔNRAS tumours with either RMC-4998 or RMC-4550 revealed downregulation of myeloid-mediated processes in RMC-4550 treated tumours relative to RMC-4998 (Supplementary Fig. 5c), which is consistent with the differences in infiltration shown in Fig. 5a. Likewise, genes encoding known myeloid-markers were depleted in tumours treated with combined RMC-4998 and RMC-4550 compared to RMC-4998 alone, after 7 days (Supplementary Fig. 5d). Along with previous reports that SHP2 inhibition can directly alter myeloid functions, these data demonstrate additional cancer-cell extrinsic properties of RMC-4550 on myeloid populations[30,36]. Additionally, we observed in tumours treated with combination of RMC-4998 and RMC-4550 increased expression of H-2K^b type I conventional dendritic cells (cDC1s), indicating that combination treatment potentially increases cross-presentation (Fig. 5e).

Besides the described changes in the myeloid populations, immunophenotyping of 3LL-ΔNRAS tumours treated with RMC-4998 or RMC-4550 also revealed an increase of the overall lymphocyte compartment, with the combination exacerbating this effect (Fig. 5f). Therefore, we next examined the impact of RMC-4998 and RMC-4550 treatment on lymphocytes, considering the role of oncogenic KRAS in

suppressing the IFN pathway, preventing subsequent antigen presentation and T cell activation, and the critical involvement of SHP2 in modulating functions of lymphocytes[37,38]. Similar to what has been previously observed with KRAS^G12C(OFF) inhibitors, inhibition of RAS^G12C(ON) led to increased tumour-infiltrating CD8^+ T and Treg cells (Fig. 6a, b and Supplementary Fig. 6a). RMC-4550-treated tumours were characterised by a further elevated presence of T cells compared to tumours treated with RMC-4998, which was also reflected in tumours treated with combined targeted therapy, again pointing towards additional effects of SHP2 inhibition on the TME independent of any effect on cancer-cell signalling (Fig. 6a, b). All treatments resulted in an increased activation of both CD8^+ and CD4^+ T cells and a shift towards an effector-memory phenotype (Fig. 6c and Supplementary Fig. 6b). Moreover, tumour-infiltrating T cells in tumours treated with both RMC-4998 and RMC-4550 showed marked activation, proliferation, and expression of potential anti-tumour cytotoxic molecules, such as TNFα, IFNγ and Granzyme B (Fig. 6d and Supplementary Fig. 6c–e). Concordant with these data, gene set enrichment analysis of tumours treated with either RMC-4550 or RMC-4998 revealed activation of IFN responses (Supplementary Fig. 6f, g), while doublet targeted therapy treated tumours showed enrichment of IFN and inflammatory-related pathways compared to either monotherapy (Fig. 6e). Likewise, there was an increase of tumour-infiltrating NK (CD49b^+Nkp46^+) cells which displayed increased proliferation in response to doublet RMC-4998 and RMC-4550 therapy, even though no differences between RMC-4998 and/or RMC-4550 treated tumours were observed in terms of NK cell numbers, indicating potential increased activity (Fig. 6f). Coinciding with this data, tumour-infiltrating NK cells in tumours treated with both RMC-4998 and RMC-4550 showed elevated expression of anti-tumour cytotoxic molecule Granzyme B (Supplementary Fig. 6h). Interestingly, tumour-infiltrating B cells (B220^+CD19^+) were uniquely increased in RMC-4550 treated tumours along with an increase of tumour-infiltrating plasma cells (CD138^+) (Fig. 6g), suggesting additional effects of SHP2 inhibition in the B cell populations.

In conclusion, these results underline a mechanistic rationale for combining RAS(ON) G12C-selective inhibitors, such as RMC-4998, with a second targeted compound, such as the SHP2 inhibitor RMC-4550, that in addition to enhancing the tumour cell intrinsic responses can directly target the TME and potentiate immune responses. Combination treatment with these compounds leads to considerable alteration of the immune TME and results in parallel activation of both innate and adaptive anti-tumour immune responses.

## RAS^G12C(ON) and SHP2 inhibition synergise with ICB in an orthotopic immune-excluded anti-PD-1 resistant model of NSCLC

As mentioned above, tissue site is an important factor for determining anti-tumour responses. Therefore, we next investigated the therapeutic impact of RMC-4998 and/or RMC-4550 in combination with

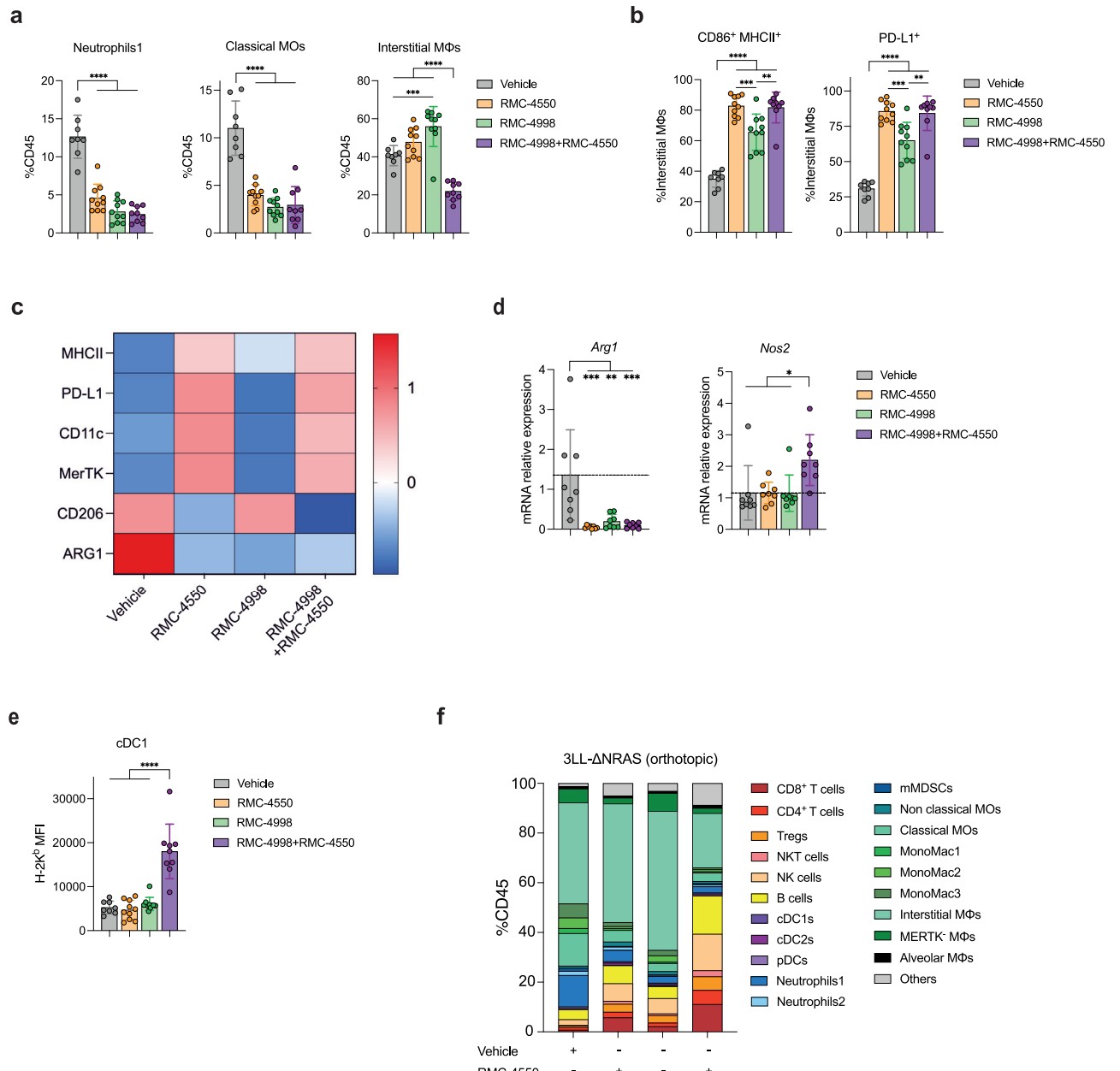

**Fig. 5 | RAS(ON) G12C-selective inhibitor RMC-4998 and SHP2 inhibitor RMC-4550 alter the myeloid cell-landscape in the TME of 3LL-ΔNRAS lung tumours.**
**a** Frequency of neutrophils, classical monocytes and interstitial macrophages in 3LL-ΔNRAS lung tumours, identified by multiparameter spectral flow cytometry, after treatment of 8 days with vehicle ($n = 8$), 100 mg/kg RMC-4998 ($n = 10$), 30 mg/kg RMC-4550 ($n = 10$) or the combination ($n = 9$). Unless otherwise stated same treatments, doses and schedule were used in the other panels. Data are mean values ± SD. Each dot represents one independent biological mouse from where isolated tumours were pooled together. Analysis was performed using one-way ANOVA test. Only significant comparisons are shown (same applies for the rest of the panels). **b** Frequency of CD86+MHCII+ (left panel) and PD-L1+ (right panel) tumour-infiltrating interstitial macrophages in treated 3LL-ΔNRAS lung tumours with vehicle ($n = 8$), RMC-4998 ($n = 10$), RMC-4550 ($n = 10$) or the combination ($n = 9$). Data are mean values ± SD. Each dot represents one independent biological mouse from where isolated tumours were pooled together. Analysis was performed using one-way ANOVA test. **c** Heatmap showing mean fluorescent intensity (MFI),

shown in Supplementary Fig. 5b, z-scores of MHCII, PD-L1, CD11c, MerTK, CD206 and ARG1 of tumour-infiltrating interstitial macrophages in treated 3LL-ΔNRAS lung tumours. Vehicle ($n = 8$), RMC-4998 ($n = 10$), RMC-4550 ($n = 10$) or the combination ($n = 9$). **d** qPCR analysis of *Arg1* and *Nos2* of 3LL-ΔNRAS lung tumours treated for 7 days with RMC-4998 and/or RMC-4550. Data are mean values ± SD; $n = 4$ mice/group. Each dot represents one tumour, 2 tumours/mouse. Analysis was performed using one-way ANOVA test. **e** Mean fluorescent intensity of H-2Kb tumour-infiltrating cDC1s in treated 3LL-ΔNRAS tumours. Data are mean values ± SD. Each dot represents one independent biological mouse from where isolated tumours were pooled together. Vehicle ($n = 8$), RMC-4998 ($n = 10$), RMC-4550 ($n = 10$) or the combination ($n = 9$). Analysis was performed using one-way ANOVA test. **f** Spectral flow cytometry immunophenotyping, as shown in Supplementary Information 1 (Supplementary Fig. 9), of 3LL-ΔNRAS lung tumours treated for 8 days with vehicle ($n = 8$), RMC-4998 ($n = 10$), RMC-4550 ($n = 10$) or the combination ($n = 9$). For all statistical analysis *$p < 0.05$, **$p < 0.01$, ***$p < 0.001$, ****$p < 0.0001$. Source data and exact $p$ values are provided as a Source Data file.

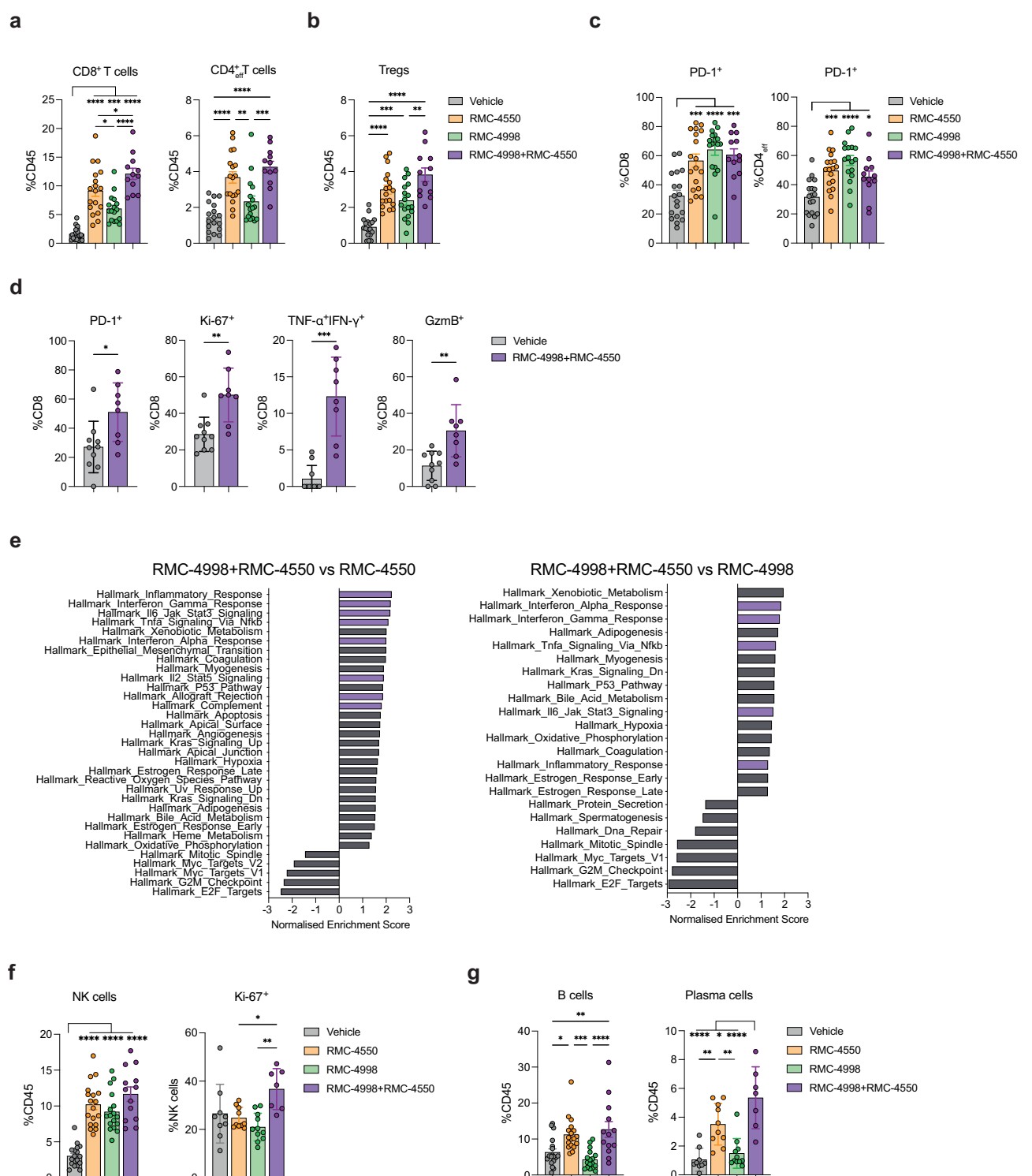

ICB therapy on the ICB-resistant 3LL-ΔNRAS model in the orthotopic lung setting. Initially, we confirmed that 3LL-ΔNRAS lung tumours are refractory to treatment with anti-PD-1, anti-CTLA-4, or the combination of both (Supplementary Fig. 7a). Next, we determined the effect of RMC-4998 and RMC-4550 on tumour growth after 7 days of treatment. Treatment with the SHP2 inhibitor RMC-4550 resulted in a slowing of tumour growth but most of the tumours were still progressing. In contrast, more than half of the tumours treated with the RAS(ON) G12C-selective inhibitor RMC-4998 regressed (Fig. 7a). Both RMC-4998 and RMC-4550 monotherapies extended survival to comparable levels, which was surprising given the lack of clear regressions induced

by RMC-4550 after 7 days of treatment (Fig. 7a, b). The long-term effects of RMC-4550 on survival may reflect the non-tumour cell intrinsic effects described earlier, which result in activation of anti-tumour responses and delay of tumour growth. In agreement with the subcutaneous data, combination of RMC-4998 and RMC-4550 enhanced tumour responses, with profound regressions in almost all the tumours analysed and an extension of survival (Fig. 7a, b).

Although treatment of tumours with either RMC-4998 or RMC-4550 monotherapy extended survival and enhanced the inflamed phenotype of the TME, addition of anti-PD-1 to either alone did not improve survival responses. In contrast, the targeted therapy doublet

**Fig. 6 | RAS(ON) G12C-selective inhibitor RMC-4998 and SHP2 inhibitor RMC-4550 induce pan lymphocytic anti-tumour response in the TME of 3LL-ΔNRAS lung tumours.** Frequency of CD8[+], CD4[+] effector T (**a**) and Treg cells (**b**) and PD-1[+] CD8[+], CD4[+] effector T cells (**c**) in 3LL-ΔNRAS orthotopic lung tumours treated for 7 days with 100 mg/kg RMC-4998, 30 mg/kg RMC-4550 or the combination. Data ± SEM of two independent experiments. Each dot represents one independent biological mouse from where isolated tumours were pooled together. Analysis was performed using one-way ANOVA test. **d** Frequency of PD-1[+], Ki-67[+], TNF-α[+]IFN-γ[+], GzmB[+] CD8[+] T cells in 3LL-ΔNRAS lung tumours treated for 7 days with combination of 100 mg/kg RMC-4998 and 30 mg/kg RMC-4550. Data are mean values ± SD. Each dot represents one independent biological mouse from where isolated tumours were pooled together. Analysis was performed using two-tailed Student's *t*-test. **e** Summary of significantly (FDR < 0.05) down- or upregulated pathways in tumours treated with combined 100 mg/kg RMC-4998 and 30 mg/kg RMC-4550 compared to RMC-4998 or RMC-4550 treated tumours (MSigDB Hallmarks). **f** Frequency of NK and Ki-67[+] NK cells in 3LL-ΔNRAS lung tumours treated for 7 days with 100 mg/kg RMC-4998, 30 mg/kg RMC-4550 or the combination. Each dot represents one independent biological mouse from where isolated tumours were pooled together. Data ± SEM of two independent experiments for NK cell frequency. Data ± SD for Ki-67[+] NK cell frequency. Analysis was performed using one-way ANOVA test. **g** Frequency of B, plasma cells in 3LL-ΔNRAS lung tumours treated for 7 days with 100 mg/kg RMC-4998, 30 mg/kg RMC-4550 or the combination. Each dot represents one independent biological mouse from where isolated tumours were pooled together. Data ± SEM of two independent experiments for B cell frequency. Data ± SD for plasma cell frequency. Analysis was performed using one-way ANOVA test. For all statistical analysis *p < 0.05, **p < 0.01, ***p < 0.001, ****p < 0.0001. Only significant combinations are shown. Source data and exact *p* values are provided as a Source Data file.

sensitised tumours to anti-PD-1, leading to significantly prolonged survival, consistent with our previous observations in subcutaneous tumours (Figs. 4c, 7b and Supplementary Fig. 4b). Micro-CT scans after 4 weeks on treatment revealed that combination of anti-PD-1 with doublet targeted therapy suppressed tumours more efficiently and prevented them from growing back (Fig. 7c). Moreover, RNA-seq of 7-day treated tumours indicated an enrichment of IFN and inflammatory-related pathways in tumours treated with combination of RMC-4998, RMC-4550 and anti-PD-1 compared to dual-targeted therapy (Fig. 7d). In agreement with these data, qPCR of treated tumours demonstrated a profound increased expression of genes involved in IFN pathway and cytotoxic response (*Ifng*, *Gzmb*, *Prf1*; Fig. 7e). Additionally, tumours treated with triple combination for 2 weeks were characterised by an elevated infiltration of CD8[+] and CD4[+] T cells (Fig. 7f). These data support the idea that combination of RMC-4998 and RMC-4550 can sensitise immune-excluded tumours to anti-PD-1 immunotherapy through enhanced IFN pathway induction and T cell persistence.

However, the triple combination still failed to generate complete responses in the orthotopic lung setting, in contrast to the subcutaneous tumours. To extend our findings, we decided to assess if the combination of RMC-4998 and RMC-4550 could sensitise tumours to other immunotherapies and generate complete responses. We therefore employed an anti-CTLA-4 antibody to attenuate CTLA-4 inhibition of positive co-stimulation by CD28. In combination with RMC-4998 and RMC-4550, anti-CTLA-4 extended survival compared with dual-targeted therapy and generated one tumour-free mouse (Fig. 7g), with 3/12 mice manifesting intestinal inflammation while the quadruple combination demonstrated the strongest effect on tumour regressions after 4 weeks of treatment (Fig. 7h) resulting in two tumour-free mice (Fig. 7g). Immunophenotyping of 3LL-ΔNRAS tumours treated for 7 days with the quadruple combination revealed increased infiltration and activation of CD8[+] and CD4[+] T cells (Supplementary Fig. 7b, c).

Several studies have demonstrated that tissue-tumour site influences anti-tumour T cell-mediated responses with T cells of the lung and draining lymph node being differentially regulated to subcutaneous setting, which could explain for differences of response observed between the two settings[34,39]. Considering these observations and that lung tumour burdened mice have more than one tumour lesion, we still manage to generate CRs in the lung setting with these combinations in this aggressive lung cancer model. These data demonstrate the potential of combining RMC-4998 and RMC-4550 to enhance tumour regressions and prolong survival in an immune-excluded, ICB-resistant NSCLC model. In parallel, this combination can be combined further with immunotherapies to generate long-term responses through enhancement of IFN responses and anti-tumour immunity.

## Discussion

The short-lived responses observed in patients treated with adagrasib and sotorasib, which target the inactive conformation of KRAS[G12C], highlight the need for the development of therapies that can prevent resistance emergence and prolong responses[8,9]. In this study, we used RMC-4998, a compound that targets the active, GTP-bound form of KRAS[G12C], and demonstrates a more rapid and potent activity compared to adagrasib. This can be attributed to the limitations of adagrasib, and other inhibitors targeting GDP-bound KRAS[G12C], which depend on the intrinsic GTPase hydrolysis rate of KRAS[G12C] to convert KRAS to the inactive state. However, while the use of active state RAS[G12C] inhibitors, as exemplified with the preclinical tool compound RMC-4998, should prevent certain resistance mechanisms, such as an increase of the active KRAS[G12C] pool[20], the present preclinical data demonstrate that adaptive resistance and MAPK reactivation, potentially caused by the feedback activation of wild-type RAS isoforms, can still be observed[14]. Here, we show that simultaneously inhibiting SHP2 using RMC-4550 suppressed MAPK reactivation, enhanced induction of apoptosis and maintained IFN pathway activation. Given the role of oncogenic KRAS in driving immune suppression, prevention of MAPK reactivation is crucial to block cancer cell-intrinsic adaptive resistance and the maintenance of an immunosuppressive TME and the potential for cross-resistance[40].

Recent studies have demonstrated the paradoxical role of the IFN pathway in cancer. We have previously shown that activation of cancer cell-intrinsic IFN response is crucial for induction of adaptive cytotoxic anti-tumour immune programmes and long-term responses induced by KRAS[G12C] inhibition, especially in combination with anti-PD-1 therapies[24]. Here, using an immunogenic lung cancer model, we also demonstrate that adaptive immunity is essential for prevention of relapse and generation of immune memory upon inhibition of KRAS and/or SHP2, highlighting the importance of adaptive immunity in generating long-term anti-tumour responses. In fact, while addition of a SHP2 inhibitor along with RAS[G12C](ON) inhibition provides a survival benefit, it is the combination of RAS[G12C](ON) inhibition with anti-PD-1 blockade that is the major driver for tumour eradication in preclinical models with a degree of intrinsic sensitivity to ICB, confirming the critical role of adaptive immunity in the generation of durable responses. However, in tumours with reduced sensitivity to KRAS[G12C] inhibition, such as the KPAR.M7 model we generated, we observed that despite being sensitised to anti-PD-1 blockade with RAS[G12C](ON) inhibition, addition of SHP2 inhibitor can enhance further anti-tumour immunity and result in tumour eradication.

We have previously shown that tumours resistant to ICB do not gain any advantage from the combination of adagrasib and anti-PD-1, even though KRAS[G12C] inhibition can partially reverse immune suppression mechanisms in these models[24]. Similarly, we do not observe an extension of survival when combining the RAS(ON) G12C-selective inhibitor RMC-4998 with anti-PD-1 in 3LL-ΔNRAS tumours, an immune-excluded model representative of lung tumours that are intrinsically resistant to ICB. In contrast, in this preclinical model the combination of KRAS(ON) G12C-selective and SHP2 inhibitors significantly extends survival, suggesting that patients with immune evasive tumours may

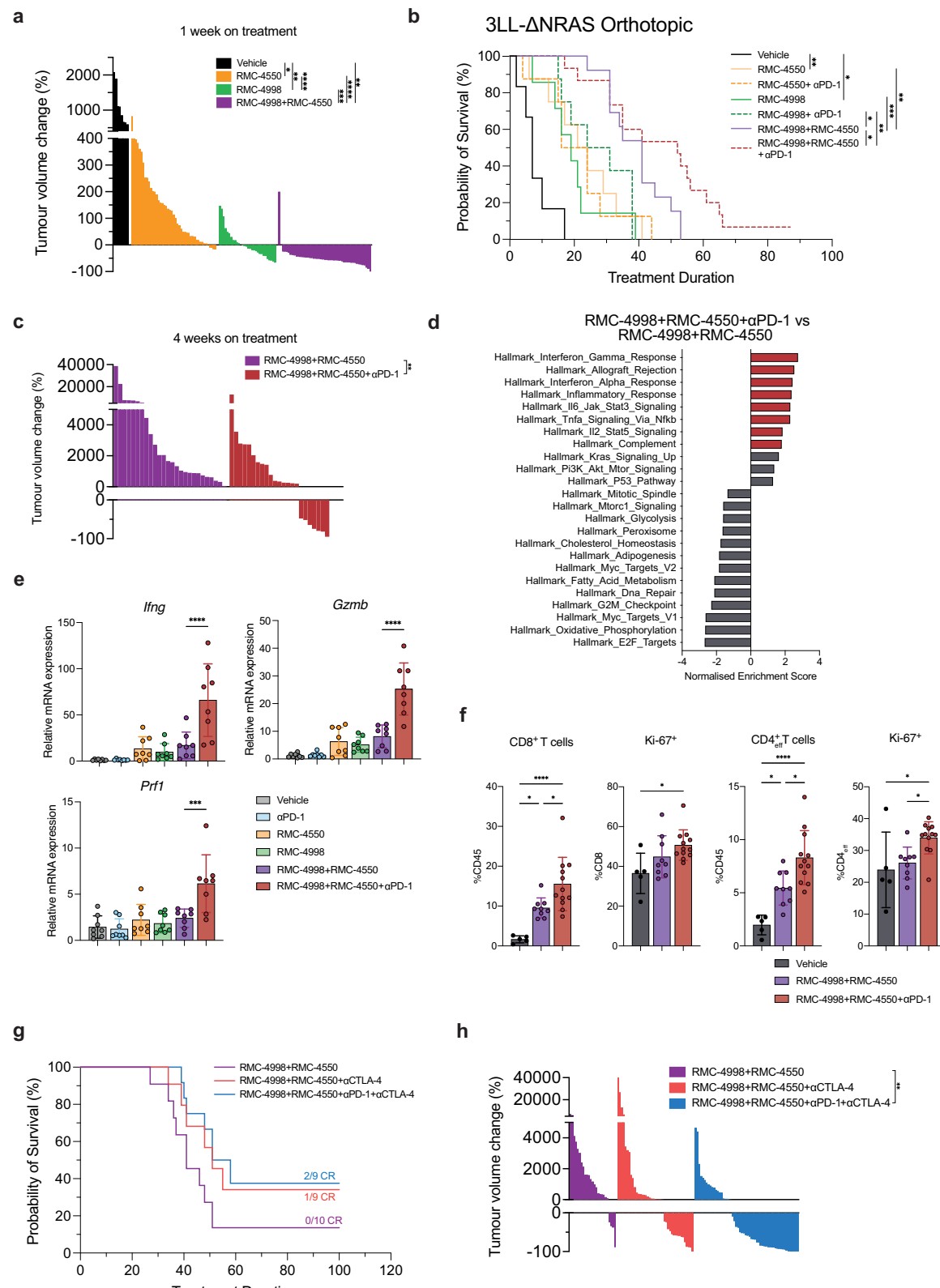

obtain greater benefits from this combination. However, indicative of the immunosuppressive nature of this model and in contrast to the immunogenic KPAR$^{G12C}$ model, combined inhibition of both KRAS$^{G12C}$ and SHP2 did not lead to tumour eradication, just delayed progression, suggestive of inadequate engagement of adaptive immunity. Nevertheless, when anti-PD-1 is added in this context, it leads to tumour eradication in some mice and development of immune memory,

indicating that targeting both KRAS$^{G12C}$ and SHP2 can sensitise non-immunogenic tumours to anti-PD-1 therapies and achieve long-term responses. In terms of response to ICB, combination treatment with RAS(ON) G12C-selective and SHP2 inhibitors appears to convert immune-excluded, cold tumours to an immune inflamed, hot, state. A noteworthy observation is that the combinations using the approved KRAS$^{G12C}$(OFF) inhibitor adagrasib can have similar effects on

**Fig. 7 | RAS(ON) G12C-selective inhibitor RMC-4998 and SHP2 inhibitor RMC-4550 synergise with ICB in an orthotopic immune-excluded anti-PD-1 resistant model of NSCLC. a** Tumour volume change after 7 days of treatment of 3LL-ΔNRAS tumours with vehicle ($n = 5$), 100 mg/kg RMC-4998 ($n = 7$), 30 mg/kg RMC-4550 ($n = 8$) or the combination ($n = 8$). Each bar represents one tumour. Analysis was done using Welch's one-way ANOVA. **b** Survival of mice bearing 3LL-ΔNRAS orthotopic lung tumours treated daily with vehicle ($n = 6$), RMC-4550 ($n = 8$), RMC-4998 ($n = 8$), RMC-4550 + αPD-1 ($n = 8$), RMC-4998 + αPD-1 ($n = 8$), RMC-4998 + RMC-4550 ($n = 13$), RMC-4998 + RMC-4550 + αPD-1 ($n = 15$). Analysis was done using log-rank Mantel–Cox test. **c** Tumour volume change of mice in (**b**) after of treatment of 3LL-ΔNRAS tumours for 4 weeks. Each bar represents one tumour. Statistics were calculated using two-tailed Mann–Whitney test. **d** Summary of significantly (FDR < 0.05) down- or upregulated pathways in 3LL-ΔNRAS tumours treated for 7 days with either the triple combination ($n = 4$ mice, 2 individual tumours/mouse) or the doublet combination ($n = 4$ mice, 2 individual tumours/mouse) (MSigDB Hallmarks). **e** qPCR analysis of 3LL-ΔNRAS orthotopic lung tumours treated for 7 days. Data are mean values ± SD; $n = 4$ mice/group. Each dot represents one tumour, 2 tumours/mouse. Analysis was calculated using one-way ANOVA. **f** Frequency of CD8$^+$ and CD4$^+$ T cells and frequency of Ki-67$^+$ CD8$^+$ and CD4$^+$ T cells in 3LL-ΔNRAS tumours after 2 weeks of treatment with vehicle ($n = 5$), RMC-4998 + RMC-4550 ($n = 9$) or RMC-4998 + RMC-4550 + anti-PD-1 ($n = 12$). Data are mean values ± SD. Each dot represents one independent biological mouse from where isolated tumours were pooled together. Analysis was calculated using one-way ANOVA. **g** Survival of mice bearing 3LL-ΔNRAS orthotopic lung tumours treated daily with the combination RMC-4998 and RMC-4550 ($n = 11$) in presence or absence of 10 mg/kg anti-CTLA-4 ($n = 12$) and 10 mg/kg anti-PD-1 ($n = 12$) (twice weekly for 2 weeks). CRs, where no tumour was detected by micro-CT scan 1 month after treatment withdrawal. **h** Tumour volume change of mice in (**g**) after 4 weeks of treatment with the combination RMC-4998 and RMC-4550 ($n = 10$) in presence or absence of anti-CTLA-4 ($n = 12$) and anti-PD-1 ($n = 12$). Each bar represents one tumour. Analysis was done using Welch's one-way ANOVA. For all statistical analysis *$p < 0.05$, **$p < 0.01$, ***$p < 0.001$, ****$p < 0.0001$. Source data and exact $p$ values are provided as a Source Data file.

immune-excluded tumours, suggesting the potential use of these combinations with the multiple KRAS$^{G12C}$(OFF) inhibitors that are currently in clinical trials. Furthermore, it is intriguing to consider whether other RAS mutant-selective inhibitors would have similar effects when combined with SHP2 inhibitors and anti-PD-1 blockade.

Given the pleiotropic role of both oncogenic KRAS and SHP2 in modulating cancer-cell intrinsic mechanisms, including indirect regulation of the TME, and the cancer-cell extrinsic functions of SHP2, there are several potential mechanisms by which combined inhibition of KRAS$^{G12C}$ and SHP2 could sensitise immune evasive tumours to anti-PD-1 therapies. In cancer cells, combination of RAS(ON) G12C-selective and SHP2 inhibitors prevents MAPK reactivation, which extends the reversion of immune evasive mechanisms driven by oncogenic KRAS, such as secretion of immune evasive cytokines and inhibition of IFN responses. Moreover, the combination results in elevated cancer cell death which could lead to increased uptake of immunogenic antigens due to phagocytosis by antigen-presenting cells[41,42]. The increase of antigen presentation along with persistent and increased IFN signalling could prime T cell responses and thus increase the dependency on the anti-PD-1/PD-L1 axis[43]. In parallel, enhanced tumour regressions and prevention of relapse could expand the T-cell reinvigoration window[44]. Alternatively, cancer-cell independent effects of SHP2 inhibition on the TME could also contribute to the sensitivity to anti-PD-1. Consistent with our data, it has been shown that SHP2 inhibition depletes immunosuppressive tumour-associated macrophages, which consequently enables recruitment of macrophages with the potential of promoting anti-tumour immunity[30]. We also demonstrate that SHP2 inhibition can lead to induction of maturation and antigen-presenting programmes in macrophages independent of cancer-cell intrinsic signalling which coincide with recent studies that have characterised the effects of SHP2 inhibition in macrophages which can lead to anti-tumour immune responses. These include induction of CXCL9 expression, a strong prognostic biomarker in several tumours, and differentiation of bone marrow-derived myeloid cells, both of which can induce CD8$^+$ T cell recruitment and activation[36,45,46]. SHP2 cancer-cell extrinsic effects are not restricted to myeloid cells. In fact, we have observed increased T cell tumour infiltration upon SHP2 inhibition compared to KRAS$^{G12C}$ inhibition. This indicates a direct role of SHP2 in T cells in terms of proliferation and/or recruitment, in agreement with previous reports[47]. Additionally, we observe increased NK cell expansion and cytotoxicity in tumours treated with doublet therapy. Niogret et al. have shown that SHP2 is indispensable for IL-15R-mediated expansion and activation of NK cells[37]. In this study, we present evidence that SHP2 inhibition can promote increased frequency of tumour-infiltrating B cells and plasma cells. B cells have been described to have a crucial role in modulating anti-tumour immunity, including

antigen presentation and antibody secretion against tumour-derived retroviral elements[48,49].

Anti-PD-(L)1 blocking antibodies, either as single therapy or in combination with chemotherapy, constitute the first line treatment for the majority of patients with KRAS-mutant NSCLC[29]. However, recent studies have also suggested the potential benefits of other immunotherapies, including anti-CTLA-4 blocking antibodies[50,51]. In this study, we demonstrate that the combination of RAS$^{G12C}$(ON) and SHP2 inhibition also sensitises immune evasive tumours to anti-CTLA-4 treatment, leading to durable responses. Interestingly, anti-CTLA-4 treatment outperforms anti-PD-1 blockade in subcutaneous tumours. Based on immunophenotyping tumours, we speculate that this differential effect may, in part, be attributed to the capacity of anti-CTLA-4 blocking the CTLA-4 immunosuppressive properties such as attenuation of CD28 signalling through binding of CD80/86[35,52,53]. However, recent studies have identified novel properties of anti-CTLA-4 treatment in activating anti-tumour immune responses, such as Fcγ receptor engagement-mediated myeloid activation[54]. Therefore, we cannot exclude additional TME influences by anti-CTLA-4 treatment. Given the numerous changes in immune cell populations within the TME induced by the combined inhibition of KRAS and SHP2, this provides rationale for combination with other immunotherapies.

Preliminary clinical activity data have been reported for the investigational agent RMC-6291, a RAS(ON) G12C-selective inhibitor, including evidence of clinical activity in patients with advanced KRAS$^{G12C}$ mutant NSCLC that had previously progressed on inactive-state KRAS$^{G12C}$ inhibitors[22]. It is still too early to know how the duration of the responses to this new generation of RAS(ON) G12C-selective inhibitors will compare to those of the first generation of KRAS$^{G12C}$(OFF) inhibitors. However, based on previous experience with targeted therapies, it is likely that resistance will eventually develop and that combinations will be needed. Combinations that maintain or even enhance the positive effects that the mutant-specific KRAS inhibitors have in the TME are of particular interest. Here we show that for immune evasive tumours, this can be achieved with additional targeting of SHP2 and that it is the combination of both the tumour cell-intrinsic and cell-extrinsic effects of SHP2 inhibition that sensitises tumours to ICB and generates durable responses. It is important to consider the potential toxicities of these combinations. Initial reports from clinical trials of sotorasib combined with anti-PD-1 checkpoint blockade have indicated high incidence of toxicities, in particular immune-related hepatoxicity[55,56]. The mechanism(s) underlying these toxicities is still unclear and requires further investigation. However, emergent data with other KRAS$^{G12C}$(OFF) inhibitors suggests that the effects are likely to reflect compound specific, off target effects, thus allowing hope that combinations involving future generations of KRAS

inhibitors could potentially be less toxic. In parallel, studies that analyse the potential of different treatment schedules to maximise the effects of the different therapies reducing the toxicities should be performed. Our studies of combination therapies targeting RAS$^{G12C}$(ON), SHP2 and immune checkpoints in preclinical models support clinical evaluation to assess their potential for improving clinical outcomes in lung cancer, providing the challenges of avoiding adverse toxicities can be addressed.

## Methods

### In vivo tumour studies

All studies were performed under a UK Home Office–approved project license and in accordance with institutional welfare guidelines. All study plans were approved by Biological Research Facility at the Francis Crick Institute. All transplantation animal experiments were carried out using 8–10-week C57BL/6J mice. Mice housing is maintained with a 12–12 h light–dark cycle and in specific-pathogen-free conditions, and each cage contains nesting and is individually ventilated and never exceeds five mice. Humidity and temperature are maintained according to UK Home Office guidelines, 20–24 °C and 45–65%, respectively.

For subcutaneous tumour injections, 150,000 KPAR$^{G12C}$, 150,000 KPAR.M7, 400,000 3LL-ΔNRAS and 200,000 KPB6$^{G12C}$ cells were mixed 1:1 with Geltrex LDEV-Free Reduced Growth Factor matrix (Thermo Fisher Scientific) and injected subcutaneously into one flank. Tumour growth was followed two or three times a week by calliper measurements and volume was calculated using the formula $0.5 \times$ (length $\times$ width$^2$). Mice were euthanised when average tumour diameter exceeded 1.5 cm. Diameter was measured using callipers. For rechallenge experiments, mice with undetectable tumours at least 30 days after the treatment withdrawn were injected subcutaneously with the same number of cells in the opposite flank. Naïve mice of similar age were injected as control.

For orthotopic tumours, 150,000 KPAR$^{G12C}$ or 10$^6$ 3LL-ΔNRAS cells were injected in 100 μl of phosphate-buffered saline (PBS) in the tail vein. Tumour volume was measured by micro-CT analysis. Mice were anaesthetised by inhalation of isoflurane and scanned using the Quantum GX2 micro-CT imaging system (PerkinElmer). Serial lung images were reconstructed and tumour volumes analysed using Analyse (AnalyzeDirect) as previously described[57]. Mice were randomised based on tumour burden, as lungs contained multiple tumours with varying numbers and size between mice. Mice were weighed and monitored regularly and were euthanized when the humane endpoint of 15% weight loss from baseline was reached or any sign of distress was observed (i.e. hunched, piloerection, difficulty of breathing). In addition, if a mouse was observed to have a tumour burden in excess of 70% of lung volume when assessed by micro-CT scanning, they were deemed at risk of rapid deterioration in health and euthanised immediately.

Mice were randomised into groups and treatments were initiated once tumours reached an average volume of 50–150 mm$^3$ for subcutaneous studies or were detectable by micro-CT for orthotopic experiments. Mice were treated daily via oral gavage with 100 mg/kg RMC-4998 and/or 30 mg/kg RMC-4550 or 100 mg/kg MRTX849. RMC-4998 was provided by Revolution Medicines, Inc. under a collaboration research agreement. RMC-4550 was provided by Revolution Medicines, Inc. under a former collaboration agreement with Sanofi. Compounds were prepared using a formulation made of 10% DMSO/20% PEG 400/10% Kolliphor HS15 in 50 mM sodium citrate buffer pH 4. MRTX849 (MedChemExpress) was prepared in 10% captisol diluted in 50 mM citrate buffer pH 5.0. For ICB treatments, mice were administered with 10 mg/kg anti-PD-1 (clone RMP1-14, cat. BE0146, BioXcell) and/or 5 mg/kg anti-CTLA-4 (clone 9H10, cat. BE0131, BioXcell) or the corresponding IgG controls (rat IgG2a and polyclonal Syrian hamster IgG, respectively). Antibodies were dissolved in PBS and administered via intraperitoneal injection (4 μl/g) twice weekly for

2 weeks. For 3LL-ΔNRAS orthotopic experiments, mice were treated with 10 mg/kg anti-PD-1 (clone RMP1-14, cat. mpd1-mab15, InvivoGen) or 10 mg/kg anti-CTLA-4 (clone 9D9, cat. mctla4-mab10, InvivoGen) or the corresponding IgG controls (Murine IgG1e3 and murine IgG2a, respectively). Antibodies were administered twice weekly for a maximum of 4 weeks.

### Sex as a biological variable

The KPAR$^{G12C}$ cell line was generated from a female mouse whereas the 3LL-ΔNRAS cell line was derived from a male mouse. Our study examined anti-tumour immune responses generated by targeted therapy and to avoid introducing error due to the induction of immune responses in female mice against genes found in the Y chromosome, we used mice with the same sex as the sex of the mouse that the cell lines were derived from, i.e., male mice for transplantation of 3LL-ΔNRAS cells and female mice for transplantation of KPAR$^{G12C}$ cells. Our study therefore examined both male and female animals, but in different settings, so it is unknown whether the specific findings are relevant for the opposite sex.

### Cell lines and treatments

NCI-H23 and Calu-1 were obtained from the Francis Crick Institute. 3LL-ΔNRAS cells were generated as previously described[58]. KPAR1.3 G12C cells (herein KPAR$^{G12C}$) and KPB6$^{G12C}$ were generated as previously described[26]. NCI-H23 cells were grown in RPMI and the rest of the cells in DMEM. Medium was supplemented with 10% foetal calf serum, 4 mM L-glutamine, penicillin (100 U/ml) and streptomycin (100 mg/ml). Cell lines were routinely tested for mycoplasma and were authenticated by short-tandem repeat DNA profiling by the Francis Crick Institute Cell Services facility.

Cells were plated at an appropriate density and left to grow for at least 24 h before treatment. MRTX849 was obtained from MedChemExpress. Recombinant mouse IFNγ was obtained from Biolegend.

KPAR.M7 was derived from orthotopic KPAR$^{G12C}$ tumours treated with 50 mg/kg MRTX849 daily 5 days a week for 6 weeks. Tumours growing during treatment, detected by micro-CT scanning, were harvested and dissociated into small pieces into a petri dish to generate cell lines which were further passaged in the culture with DMEM-F12 medium supplemented with GlutaMAX®, 10% FBS, 1 μM hydrocortisone, 20 ng/ml mEGF, 50 ng/ml mIGF, penicillin (100 U/ml) and streptomycin (100 mg/ml) and 50 nM of MRTX849, which was removed from the media after three passages. Cell lines were single sorted into a 96-well plate and single clones were expanded to generate KPAR.M7 cell line (has not yet been authenticated).

### Cell viability and apoptosis assays

For viability assays, cells were grown in 96-well plates and inhibitors were added 24 h later. After 72 h, 5 μl of CellTiter-Blue (Promega) was added, and cells were incubated for 90 min at 37 °C before measuring fluorescence using an EnVision plate reader (PerkinElmer). For longer-term proliferation assays, cells were plated in 24-well plates and treated for 6 days. Compounds were replaced after 3 days of treatment. At the end of treatment, cells were fixed and stained using 0.2% crystal violet in 2% ethanol.

Percentage of apoptotic cells was measured using flow cytometry. Cells were plated in 6-well plates and treated for 72 h. At the end of the treatment, harvested cells were resuspended in Annexin V binding buffer and stained with FITC Annexin V (BD Biosciences) and DAPI.

### Western blotting

Cells were plated in 6-well plates and treated 24 h later. At the end of the treatment, cells were lysed using 10X Cell Lysis Buffer (Cell Signalling) supplemented with Complete Mini protease inhibitor cocktail and PhosSTOP phosphatase inhibitors (Roche). After quantification using a BCA protein assay kit (Pierce), proteins (15–20 μg) were

separated on 4–12% NuPAGE Bis–Tris gels (Life Technologies) followed by transfer to PVDF membranes. Bound primary antibodies were incubated with HRP-conjugated secondary antibodies (Amersham) and detected using chemiluminescence (Luminata HRP substrate, Millipore). List of antibodies used and scanned blots can be found in Supplementary Information, Supplementary Table 2.

## Quantitative RT-PCR
RNA was extracted from cell lines or frozen lung tumours using the RNeasy Mini Kit (Qiagen) following the manufacturer's instructions. For in vivo tumour samples, tumours individually isolated from lungs were lysed and homogenised using RNase-free disposable pellet pestles (Kimble Chase) followed by QIAshredder columns (Qiagen). cDNA was generated using the Maxima First Strand cDNA Synthesis Kit (Thermo Fisher Scientific) and qPCR was performed using Fast SYBR Green Master Mix (Applied Biosystems). List of primers used is detailed in Supplementary Information, Supplementary Table 4. Gene expression changes relative to the housekeeping genes were calculated using the ΔΔCT method.

## RNA sequencing
RNA was extracted as indicated above. RNA quality was measured using the TapeStation 4000 (Agilent). Libraries were prepared using the NEBNext Ultra II Directional PolyA mRNA (New England Biolabs) according to manufacturer's instructions, with an input of 150 ng and 10 PCR cycles for library amplification. Library quality was measured using the TapeStation 4200 (Agilent). Samples were sequenced to a depth of ~25M paired-end 100 bp reads in an Illumina NovaSeq 6000 system.

For data analysis, sequencing reads were pre-processed and mapped to the mouse GRCm38 genome using the nfcore/rnaseq pipeline with STAR and RSEM. The R package DESeq2 was used to perform differential analysis between treatment groups. Gene set enrichment analysis was carried out with the R package fgsea using a ranked list of Wald statistic values calculated by DESeq2 for each comparison. Genes were taken from the Hallmark set available at MSigDB using the R package msigdbr.

## Spectral and conventional flow cytometry
Mice were euthanised using schedule 1 methods, lung tumours were isolated and all tumours from one lung were pooled together. Tumours were finely cut into small pieces and digested with collagenase (1 mg/ml; Thermo Fisher Scientific) and DNase I (50 U/ml; Life Technologies) in HBSS for 45 min at 37 °C. Samples were filtered through 70 µm strainers (Falcon) and red blood cells were lysed using ACK buffer (Life Technologies). For conventional flow cytometry, after washes in PBS, cells were stained with fixable viability dye eFluor870 (BD Horizon) for 30 min and blocked with CD16/32 antibody (BioLegend) for 10 min. Samples were stained using fluorescently labelled antibody mixes for surface markers and then washed three times in FACS buffer (2 mM EDTA and 0.5% bovine serum albumin in PBS, pH 7.2). For spectral flow cytometry cells were blocked with CD16/32 antibody (BioLegend) for 5 min at 4 °C and then stained at room temperature along with fixable Zombie NIR (Biolegend) viability dye. After staining, samples were fixed in Fix/lyse solution (eBioscience). If intracellular staining was performed, cells were instead fixed and permeabilized with Foxp3/Transcription Factor Fixation/Permeabilization Kit (Invitrogen) according to manufacturer's instructions before staining with intracellular antibodies. Single stain controls with spleen or OneComp eBeads (Invitrogen) for conventional and UltraComp eBeads (Invitrogen) for spectral flow cytometry were performed. List of antibodies used is detailed in Supplementary Information, Supplementary Tables 1 and 3. Samples were resuspended in FACS buffer and analysed using a FACSymphony cytometer (BD) for conventional and

5L Spectral Analyser Aurora (Cytek) for spectral flow cytometry. Data were analysed using FlowJo software, gating strategies are detailed in Supplementary Fig. 8 for conventional and Supplementary Fig. 9 for spectral flow cytometry. For intracellular staining of NK and T cells, cell suspensions were incubated in RPMI supplemented with 1:1000 BD GolgiPlug (BD Biosciences), 1 µg/ml ionomycin, 50 ng/ml PMA (all Sigma) for 1 and 4 h, respectively, and then staining was performed using the Foxp3/Transcription Factor Fixation/Permeabilization Kit (Invitrogen).

## Multiplex immunofluorescence
Samples were fixed for 24 h in 10% NBF before transferring to 70% EtOH. Paraffin embedding of samples was carried out using Tissue-Tek VIP® 6 AI processor. Three micrometres of FFPE sections were cut and baked for 1 h at 60 °C before staining was performed on the BondRx Leica Bond Rx platform. Hydrogen peroxide (3%) was used to block the endogenous peroxidase and 0.1% BSA solution was used for protein blocking. Antigen retrieval stripping steps between each antibody were performed with either Epitope Retrieval Solution 1 or Epitope Retrieval Solution 2 for 20 min. Triple IF staining was performed and antibodies were applied with Opal™ pairings in the following order: FoxP3 (CST, 12653) 1:400 with Opal 570 1:500, CD8 (Abcam, ab217344) 1:500 with Opal 690 1:200 and CD4 (Abcam, ab183685) 1:750 with Opal 520 1:500. Bond anti-rabbit Polymer (Leica, Novolink Max Polymer, RE7260-CE) was used as secondary for all antibodies. Slides were counterstained with DAPI (Thermo Scientific, 62248) 1:2500. Slides were mounted with Prolong Gold Antifade Reagent (Invitrogen, P36934) and scanned using the PhenoImager HT (formerly Vectra Polaris) using MOTiF scanning mode.

For 8-plex immunofluorescence antibodies were applied with Opal™ pairings in the following order: arginase-1 (CST, 93668) 1:500 with Opal 690 1:150, PD-1 (Abcam, ab214421) 1:500 with Opal 620 1:200, B220 (BD Biosciences, 553086) 1:750 with Opal 520 1:300, CD4 (Abcam, ab183685) 1:750 with Opal 480 1:150, Ki67 (Abcam, ab15580) 1:2500 with Opal 570 1:300, cCasp3 (CST, 9579S) 1:500 with Opal 650 1:700, CD8 (Abcam, ab217344) 1:500 with TSA-DIG 1:100 followed by anti-DIG-Opal 780 1:25. Bond anti-rabbit Polymer (Leica, RE7260-CE) was used as secondary for antibodies raised in rabbit. Streptavidin-peroxidase (Dako, P0397) 1:500 was used for biotinylated antibody raised in rat. Slides were counterstained with DAPI (Thermo Scientific, 62248) 1:2500. Slides were mounted with Prolong Gold Antifade Reagent (Invitrogen, P36934) and scanned using the PhenoImager HT (formerly Vectra Polaris) using Field scanning mode. Tissue-specific spectral library was produced and spectral unmixing with autofluorescence removal was performed using Phenochart™ and Inform® software. Segmentation and protein analysis was done using QuPath 0.5.0x64.

## Statistical analysis
Data were analysed using Prism 8 (GraphPad Software) using normality distribution tests, Mantel–Cox, two-tailed Student's t-test, one-way, two-way or Welch's ANOVA, as indicated. Significance was determined at $p < 0.05$ (*$p < 0.05$, **$p < 0.01$, ***$p < 0.001$, ****$p < 0.0001$).

## Reporting summary
Further information on research design is available in the Nature Portfolio Reporting Summary linked to this article.

# Data availability
The RNA-seq data have been deposited in the Gene Expression Omnibus database under accession number GSE254755. The remaining data are available within the article, Supplementary Information or Source Data file. Source data are provided with this paper.

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

## Acknowledgements

We thank the science technology platforms at the Francis Crick Institute including Biological Resources, Advanced Sequencing, Scientific Computing, Bioinformatics and Biostatistics, Flow Cytometry, Experimental Histopathology and Cell Services. We thank the members of the Oncogene Biology Laboratory for their discussions and critical reading of the manuscript. This work was supported by the Francis Crick Institute which receives its core funding from Cancer Research UK (FC001070), the UK Medical Research Council (FC001070) and the Wellcome Trust (FC001070). This work also received funding from the European Research Council Advanced Grant RASImmune, from a Wellcome Trust Senior Investigator Award 103799/Z/14/Z, and from Revolution Medicines, Inc. under a collaborative research agreement.

## Author contributions

P.A., M.M.-A. and J.D. designed the study, interpreted the results and wrote the manuscript. P.A., S.R., M.T., C.P., A.dC., J.B., E.M., S.C.T. and A.M. performed the biochemical experiments. C.M. assisted with in vivo studies. R.G. performed bioinformatic analysis. C.B., E.Q. and J.A.M.S. contributed with interpretation and resources. All authors contributed to the manuscript revision and review.

## Funding

## Competing interests

J.D. has acted as a consultant for AstraZeneca, Jubilant, Theras, Roche and Vividion and has funded research agreements with Bristol Myers Squibb, Revolution Medicines and AstraZeneca. S.C.T. has acted as a consultant for Revolution Medicines. C.B., E.Q. and J.A.M.S. are employees of Revolution Medicines. The other authors declare that they have no competing interests.
