## [Peer Review File · Nature Communications]

Combining RAS(ON) G12C-selective inhibitor with SHP2 inhibition sensitises lung tumours to immune checkpoint blockadeREVIEWER COMMENTS

Reviewer #1 (Remarks to the Author): with expertise in lung cancer, (immuno)therapy

In this manuscript, Anastasiou et al., interrogate the efficacy and immune sensitizing effects of the GTP-bound RASG12C inhibitor RMC-4998 in combination with the SHP2 inhibitor RMC-4550, with or without ICI in the context of either T-cell inflamed (KPARG12C) or immune excluded (3LL-ΔNRAS) models of lung cancer. Overall the study is well conducted, the group has significant expertise in the intersection of RAS-targeted therapies and immunotherapies and the experiments are straightforward and follow a similar path to the author's previously published work. Elements of the work are novel, although others have been previously described in the literature, albeit not specifically in the context of the RMC-4998 plus RMC-4550 combination. A number of limitations and concerns mainly related to the breadth and scope of the models used and thus the generalizability of conclusions as well as the at times incomplete mechanistic studies also arise and reduce enthusiasm for the manuscript in its current form. Nonetheless these caveats should be addressable within a reasonable time frame. Below I outline key critiques:

1. Since effects on OFF state selective inhibitors with SHP2 inhibitors have been previously reported one important question at the core of this study is whether combinations of RMC-4998 with RMC-4550 exhibit favorable efficacy or immune modulatory effects compared with OFF state selective inhibitors in combination with RMC-4550. Although adagarsib is used in in vitro studies, none of the in vivo studies include adagrasib (or sotorasib) plus RMC-4550 as a control (this would be particularly relevant for the combinations with anti-PD-1 in orthotopic tumors). This is important as several clinical trials of OFF state selective inhibitors plus SHP2 inhibitors (including the clinical compound RMC-4630) are in clinical development. Is RMC-4998 plus RMC-4550 equivalent or superior to adagrasib/sotorasib plus RMC-4550?
2. The conclusion that in hot tumors the main contribution towards long term efficacy comes from the RMC-4998 + anti-PD-1 combo with limited additional benefit from the addition of SHP2 inhibition is interesting but premature as it is based on a single model (KPARG12C) that appears quite sensitive to G12C inhibition. The authors should validate this conclusion in models with intermediate levels of KRAS G12C inhibitor sensitivity. It is possible that in this setting of incomplete MAPK inhibition with a RASG12C inhibitor alone,

addition of SHP2 will be required to induce stronger and sustained MAPK pathway suppression, immune sensitization and synergy with anti-PD-1 therapy. This distinction is important since it may impact the clinical development path of novel RAS inhibitors such as RMC-4998 in tumors with high levels of PD-L1 expression (PD-L1 tumor proportion score $\geq 50\%$).

3. Similarly, I have difficulty generalizing conclusions from a single “cold” model (especially with engineered deletion of NRAS) that does not recapitulate the full spectrum of immune cell excluded KRASG12C lung cancer to the entirety of the disease. The authors have previously described that KP tumors in the genetically engineered model (or in syngeneic transplantable models without the engineered) are cold and poorly responsive to anti-PD-1. Extending these studies to KP (and/or Kras;Lkb1 tumors that are inherently cold) tumors in the GEMM would significantly substantiate the key conclusions regarding the ability of RMC-4998 plus RMC-4550 plus anti-PD-1 to cause tumor inflammation and induce regression of cold tumors. It would also be useful to assess the impact of RMC-4998 plus RMC-4550 in models with a predominantly neutrophil-rich microenvironment (since the unique effect of SHP2 inhibition appear to be observed in the TAM population).

4. Although addition of anti-CTLA-4 appears to have a significant impact on clinical outcomes and in particular on rates of tumor eradication, this finding appears to be somewhat discounted by the authors and attributed to depletion of Tregs, which may be a mouse specific effect that is not observed in human tumors as described previously. Nonetheless this is a speculative conclusion and this needs to be tested, since the authors utilize anti-CTLA-4 combinations throughout the manuscript. At a minimum, immune profiling studies should be extended to these treatment arms. Of note, several approved first line regimens of chemo-immunotherapy in metastatic non-small cell lung cancer incorporate anti-CTLA-4 in combination with anti-PD-1/anti-PD-L1 with or without chemotherapy.

5. Additional functional studies -beyond immune profiling - would go a long way towards cementing the contribution of SHP2 inhibition to remodeling of the myeloid cell landscape. For example the authors could isolate macrophages and co-culture them with activated lymphocytes.

6. More detailed single cell RNA-Seq studies would no doubt have shed critical novel insights into sub-populations of myeloid and lymphoid cells that mediate the differential effects of

RMC-4998, RMC-4550 and their combination and would further have allowed assessment of marker expression in individual cell populations rather than in bulk tumors.

7. Is the added effect specific to CTLA-4 blockade or is it observed also with inhibitors of other immune checkpoints such as TIM-3, LAG-3 or 4-1BB agonist antibodies?

8. In the 3LL- Δ NRAS orthotopic model, the quadruple treatment arm with RMC-4998+RMC-4550+aPD-1+aCTLA-4 is missing. The argument that such a combination may not be tolerable in patients is plausible but based on the authors previous results with anti-CTLA-4 this is an important arm to include and would further have addressed the question as to whether the effect of CTLA-4 blockade is limited to the subcutaneous models.

Reviewer #2 (Remarks to the Author): with expertise in lung cancer, (immuno)therapy

The manuscript entitled “Combining RASG12C(ON) inhibitor with SHP2 inhibition sensitises immune excluded lung tumours to immune checkpoint blockade” by Anastasiou et al. investigates novel drug combinations for refractory NSCLC. Traditional KrasG12C inhibitors such as adagrasib and sotorasib bind to the GDP bound KrasG12C and keep Kras in its inactive state. These are denoted as KrasG12C (OFF) inhibitors. Resistance to these inhibitors is common when used in the clinic to treat NSCLC and tumors recur within 7 months. Feedback reactivation of upstream RTKs, which result in wt or mutant Kras activation is a common resistance mechanism and so is the increased expression of mutant Kras. Thus, new inhibitors and or combinations of new therapies are needed to prevent resistance and treat refractory NSCLC.

In this study, the authors utilized a new KrasG12C inhibitor, which is selective for the active state of RASG12C(ON). Furthermore, the authors rationalize the combination with a SHP2 inhibitor, which should prevent feedback RTK signaling thereby preventing resistance.

Lastly, the authors investigate combination of the two compounds with ICB in NSCLC mouse model of various immunogenicity. To my knowledge these combinations has not been tested and is highly relevant as these KrasG12C ON inhibitors are currently tested in clinical trials.

The manuscript is well written and logically constructed. The authors investigate single agents, combination therapy as well as sensitization to ICB. Two models are utilized, which have previously been shown to be immunogenic (KPARG12C) or immune excluded (3LL-

Δ NRAS) to test their hypotheses.

The authors conclude that relapse can be prevented with the combo in immunogenic tumors and induce an immune memory and that combo sensitizes to ICB in immune excluded 'cold' tumors by remodeling of the TME. The data is well presented and generally sound, however, since only two different models were used the question remains whether the therapeutic responses observed are dependent on differences in the TME of the two models or just cell line specific. It is unclear why expression of KrasG12C in presumably same lung epithelial cell origin results in an immunogenic or immune cold tumor. The authors show that the immune cold model is refractory to any ICB therapy and refer to previously published data about immunogenicity of the other model however detailed TME is not assessed by histology or flow cytometry prior to treatment in both models. It remains to be investigated why the murine KPARG12C is immunogenic and the 3LL- Δ NRAS is not. It is interesting that PD-1 therapy is not working in the immune excluded model as the authors show that PDL-1 expression appears in 40% of TAMs express PD-L1 in the vehicle treated? Is this because of the low number of T cells? A discussion of this would be helpful.

Nevertheless, the combination of the KrasG12C (ON) inhibitor with SHP2 inhibitor sensitizes to ICB in both models and appear to induce an adaptive immune response in both, thus the title and conclusion should be rephrased to include these findings. In general, there are several minor concerns pertaining to labeling and inclusion of statistical significance and the route of administration of all drugs/antibodies, which is omitted in Fig. legend, but found in methods.

Minor concerns:

- Figure 1A: stats are missing, which makes it difficult to conclude that efficacy of one drug is better than the other.
- Figure 1C: "combination of RMC-4550 with RMC-4998 prevented the rebound of ERK phosphorylation at later time points (24-48h)". There appears to be pERK expression at 24-48hr in the CALU1 cells, so this statement should be toned down.
- Figure 1D: there are no stats for viability graphs and thus it appears like an overinterpretation to state that the "combination of RMC-4550 with RMC-4998 prevented the rebound of ERK phosphorylation at later time points (24-48h) and resulted in a stronger reduction in cell viability".

- Figure 1E: stats are missing.
- Figure 1H: first panel, hard to read since the first two lines indicating significance are at the same height. Also, there doesn't appear to be a statistical significance of the combination when compared to single agent treatment alone. Needs to be added to the text and should be indicated as ns in graph.
- Figure legends 2-6, route of injection for drugs and ICB and dosing schedule for ICB are missing from the legend, but are listed in methods section.
- Why does tumor volume graph in Figure 2A go out 45 days but tumor volume graphs with indicated CR go out to 70 days in Figure 2B?
- Figure 2C: stats missing. Looks like the combination displayed higher activity?
- Figure 2D: stats for comparing single agent with combination are not indicated. Please add ns to the graph and mention in the result section.
- Figure 2F: CR should be separated from the survival, way too confusing and can be inserted as a table. Also, neither in Figure legend nor in methods it is indicated whether appropriate IgG control was used in the groups not receiving aPD-1.
- Figure 3A: stats are described in the fig legend but not in panel.
- Figure 3B: the lines and stars indicating significance are somewhat overlapping and hard to tell what groups are being compared.
- Figure 3B-E: The combo group with CTLA-4 alone should be included in the Figures 3B-E and stats should be added. Looking at the CTLA-4 data in Figs. Could it be that the effect is CTLA-4 dependent and not additive as implied by authors. Figures 3C-E can be moved in supplemental and instead show table with CR and stats.
- Instead of Figure 3C-E, the Figure S3c,d data could be moved into the main body.
- Figure 4C statistical significance comparisons should be the same as in Figure 4B, panel two: single agent alone compared to combo. Statistical sig. compared to vehicle could be indicated in Fig. legends.
- Figure 5A, B: comparison of single with combo should always be indicated even if ns.
- Figure 5A,B should indicate if data is from orthotopic tumors.
- Figure 6F: text line 352: "elevated infiltration and persistent proliferation of CD8+ and CD4+T cells (Fig. 6F). There is no statistical diff. between combo and combo plus ICB in CD8 T cells", please tone down the conclusion.
- Was combination of PD1 and CTLA-4 tried in subcutaneous model (Figure 6H)

- Figure 4 doesn't explain the CR observed in the different tx groups (in Figure 3) since there are very few differences between single agent and combo. Plz discuss appropriately.
- Add to discussion that the observed different outcome between the groups in tumor volume and survival (Figure 6) could also be due to the difficulty of measuring tumor volume with microCT?

Reviewer #3 (Remarks to the Author): with expertise in lung cancer, (immuno)therapy

This is an important characterization study of a new G12C inhibitor in combination with a Shp2 inhibitor as new G12C inhibitors are needed in the clinic. Study is highly rigorous. Some aspects of the study need strengthening. There are two novel drugs, many different combinations resulting in different immune phenotypes. Differences in the response of subcutaneous vs orthotopic models to the treatments are not discussed or analyzed.

One question relates to the model used in most of the experiments-KPAR-G12C, this model should be described. Subcutaneous tumors are relatively small when treatments are initiated. Volume of orthotopic tumors are not shown.

Does RMC-G12C inhibitor delays resistance as compared to sotorasib in vivo in KPARC model? Would this drug in combination with the Shp2 similarly induce anti-tumor immunity in a non-G12C driven model?

Mechanistic insights into the reasons for immune cell sculpting in both KPARC and 3LL needs strengthening. Is there a direct connection and if so, what is the connection?

Would the 4-drug combination cause enhanced toxicity in patients since Kras inhibitors in combination with pembroluzimab causes toxicities. Along these lines, in the orthotopic models in figures 1 and 6 (Kpar vs 3LL): there is not an obvious difference between RMC4998 vs RM4998+RMC4550 groups.

What is the antigen in non-immunogenic -3LL tumors? If there are antigens, then does the baseline tumor microenvironment dictate the differences observed between KPARC and 3LL with the treatments?

Authors mention changes in B and NK cells with the combination treatment. What is the role of B cells in this therapeutic response and do they contribute to the response mediated by Kras G12C and Shp2 inhibitors?

Some of the individual treatment arm tumor measurement data can be moved to supplementary figures and immune characterization of KPARC model should be moved to main figures. For the immune characterization some more detail is needed, are the different groups normalized by the tumor amount?

RESPONSE TO REVIEWERS

Nature Communications manuscript NCOMMS-24-04858, Anastasiou et al.

Reviewer comments in italics, authors' responses in regular font

Reviewer #1

In this manuscript, Anastasiou et al., interrogate the efficacy and immune sensitizing effects of the GTP-bound RASG12C inhibitor RMC-4998 in combination with the SHP2 inhibitor RMC-4550, with or without ICI in the context of either T-cell inflamed (KPARG12C) or immune excluded (3LL-ΔNRAS) models of lung cancer. Overall the study is well conducted, the group has significant expertise in the intersection of RAS-targeted therapies and immunotherapies and the experiments are straightforward and follow a similar path to the author's previously published work. Elements of the work are novel, although others have been previously described in the literature, albeit not specifically in the context of the RMC-4998 plus RMC-4550 combination. A number of limitations and concerns mainly related to the breadth and scope of the models used and thus the generalizability of conclusions as well as the at times incomplete mechanistic studies also arise and reduce enthusiasm for the manuscript in its current form. Nonetheless these caveats should be addressable within a reasonable time frame. Below I outline key critiques:

1. Since effects on OFF state selective inhibitors with SHP2 inhibitors have been previously reported one important question at the core of this study is whether combinations of RMC-4998 with RMC-4550 exhibit favorable efficacy or immune modulatory effects compared with OFF state selective inhibitors in combination with RMC-4550. Although adagrasib is used in in vitro studies, none of the in vivo studies include adagrasib (or sotorasib) plus RMC-4550 as a control (this would be particularly relevant for the combinations with anti-PD-1 in orthotopic tumors). This is important as several clinical trials of OFF state selective inhibitors plus SHP2 inhibitors (including the clinical compound RMC-4630) are in clinical development. Is RMC-4998 plus RMC-4550 equivalent or superior to adagrasib/sotorasib plus RMC-4550?

This is a very relevant question. We have compared the responses of adagrasib or RMC-4998 in combination with SHP2 inhibitor (SHP2i) or SHP2i + PD-1 in 3LL-ΔNRAS subcutaneous tumours. We observe similar responses to adagrasib + RMC-4550 compared with RMC-4998 + RMC-4550 (new Supplementary Figure 4e). Head-to-head comparisons of two drugs with closely related but not identical mechanisms of action are complicated by a wide range of factors and we would be very cautious about drawing too much by way of conclusion from these limited data. However, in these experiments, adagrasib is not inferior to RMC-4998, nor is adagrasib plus RMC-4550 inferior to RMC-4998 plus RMC-4550. This might imply that the cancer-cell independent effects of SHP2 inhibition acting on cells in the TME, especially macrophages, as first described in reference #30 (PMID: 32350067) and further elucidated in this manuscript, may be more important than the cancer cell autonomous effects of SHP2 inhibition in deepening the amplitude and duration of inhibition of KRAS signaling by KRAS G12C (OFF) inhibitor. In addition, we now show that the combination of KRAS^{G12C} inhibitor (G12Ci) and SHP2 inhibition also sensitizes tumours to anti-PD-1 and generates durable complete responses when the KRAS^{G12C}(OFF) inhibitor adagrasib is used (new Supplementary Figure 4f). This is highly relevant as it suggests that this combination could also be used with the multiple KRAS^{G12C}(OFF) inhibitors currently in clinical trials, some of them already being tested as the dual combinations of G12Ci(OFF)+SHP2i or G12Ci(OFF)+anti-PD-1. This has now been indicated in the discussion.

2. The conclusion that in hot tumors the main contribution towards long term efficacy comes from the RMC-4998 + anti-PD-1 combo with limited additional benefit from the addition of SHP2 inhibition is interesting but premature as it is based on a single model (KPARG12C) that appears quite sensitive to G12C inhibition. The authors should validate this conclusion in models with intermediate levels of KRAS G12C inhibitor sensitivity. It is possible that in this setting of incomplete MAPK inhibition with a RASG12C inhibitor alone, addition of SHP2 will be required to induce stronger and sustained MAPK pathway

suppression, immune sensitization and synergy with anti-PD-1 therapy. This distinction is important since it may impact the clinical development path of novel RAS inhibitors such as RMC-4998 in tumors with high levels of PD-L1 expression (PD-L1 tumor proportion score $\geq 50\%$).

The reviewer makes an important point. To address this question, we have developed a new cell line which is partially resistant to G12C inhibitors. This cell line has been isolated from one KPAR^{G12C} orthotopic tumour that progressed upon extended adagrasib treatment; details have been added in the Methods section, the line is termed KPAR.M7. As the reviewer hypothesised, using a less sensitive cell line we now observe an additional benefit from the addition of SHP2 inhibitor to the combination of RMC-4998 + anti-PD-1. In fact, we only generated durable complete regressions when using the triple drug combination and not with either double drug combination. These experimental data have been added in the new Figures 3e, 3f and S3b. These results strengthen the importance of the triple combination for the treatment of those tumours with partial responses to KRAS^{G12C} inhibition and/or anti-PD-1 therapies.

3. Similarly, I have difficulty generalizing conclusions from a single “cold” model (especially with engineered deletion of NRAS) that does not recapitulate the full spectrum of immune cell excluded KRASG12C lung cancer to the entirety of the disease. The authors have previously described that KP tumors in the genetically engineered model (or in syngeneic transplantable models without the engineered) are cold and poorly responsive to anti-PD-1. Extending these studies to KP (and/or Kras;Lkb1 tumors that are inherently cold) tumors in the GEMM would significantly substantiate the key conclusions regarding the ability of RMC-4998 plus RMC-4550 plus anti-PD-1 to cause tumor inflammation and induce regression of cold tumors. It would also be useful to assess the impact of RMC-4998 plus RMC-4550 in models with a predominantly neutrophil-rich microenvironment (since the unique effect of SHP2 inhibition appear to be observed in the TAM population).

To address this comment, we have used a syngeneic transplantable model derived from KP mice (KPB6^{G12C} cells). We have previously shown that KPB6 tumours are significantly infiltrated with neutrophils and show low infiltration of T cells (PMID: 35930804, reference #26 here). Although treatment with G12Ci increases T cell infiltration, this is not enough to sensitise these tumours to anti-PD-1 treatment (PMID: 35857848, reference #24 here). New Figure 4f shows that the addition of SHP2i to the combination with G12C(ON) inhibitor sensitises tumours to anti-PD-1 and enhances tumour growth inhibition, which validates the benefit of the triple combination in another model. However, the combination is not able to generate complete regressions in this model, probably due to the low number of clonal tumour neoantigens.

4. Although addition of anti-CTLA-4 appears to have a significant impact on clinical outcomes and in particular on rates of tumor eradication, this finding appears to be somewhat discounted by the authors and attributed to depletion of Tregs, which may be a mouse specific effect that is not observed in human tumors as described previously. Nonetheless this is a speculative conclusion and this needs to be tested, since the authors utilize anti-CTLA-4 combinations throughout the manuscript. At a minimum, immune profiling studies should be extended to these treatment arms. Of note, several approved first line regimens of chemo-immunotherapy in metastatic non-small cell lung cancer incorporate anti-CTLA-4 in combination with anti-PD-1/anti-PD-L1 with or without chemotherapy.

The initial hypothesis that the mechanism of action of anti-CTLA-4 antibodies being in part due to depletion of regulatory T cells was based on our previous data in mouse models showing that CTLA-4 treatment depletes Tregs in subcutaneous tumours (PMID: 35930804, reference #26 here) and to current data from our lab in orthotopic tumours (Cole et al., bioRxiv 2024, doi: <https://doi.org/10.1101/2024.04.11.588725>). The anti-CTLA-4 antibodies are murinised IgG2a, so would be expected to deplete cells expressing CTLA-4. To try to provide more evidence for this, as the reviewer suggested, we have extended the immune profiling analysis to the arms with CTLA-4. New Supplementary Figure 7 shows that this anti-CTLA-4 antibody boosts the levels of conventional CD4 and CD8 T cells when added to the KRASi plus SHP2i combination. Addition of anti-CTLA-4 does lead to decreases in the Treg population, but these fall short of

statistical significance: we have therefore removed the attribution of the effect of CTLA-4 to the impact on Tregs from the orthotopic results and the discussion sections.

5. Additional functional studies -beyond immune profiling - would go a long way towards cementing the contribution of SHP2 inhibition to remodeling of the myeloid cell landscape. For example the authors could isolate macrophages and co-culture them with activated lymphocytes.

These are very interesting experiments. It has already been shown that the SHP2 inhibitor RMC-4550 drives selective depletion of pro-tumorigenic M2-like macrophages. Moreover, functional assays co-culturing human MDSC and T cells showed that RMC-4550 alone had no effect on T-cell proliferation or cytokine release but was able to block the antiproliferative effects of MDSCs on CD8⁺ T cells (PMID: 32350067, reference #30 here). However, these studies might not reflex the complexity of the tumour microenvironment and heterogeneity of the macrophage population. To characterise better the effects of SHP2i and G12C(ON)i in the myeloid populations, we have deepened our analysis designing a panel of 36 markers for multiparametric spectral flow cytometry (details in Methods and new Supplementary Figure 9). New data are shown in Figure 5b, c, f and S5b. We show that SHP2i leads to a greater proportion of CD86⁺ MHCII⁺ and PD-L1⁺ interstitial macrophages compared to G12C(ON)i, suggesting that some effects driven by SHP2 inhibition are independent of the cancer-cell intrinsic KRAS signalling. By incorporating more markers associated with myeloid function we show that SHP2i, but not G12C(ON)i, induces maturation and antigen presentation. These enhanced effects of SHP2i in the interstitial macrophage population are correlated with the increase in T cell infiltration (Fig. 6a) again reinforcing the suggestion that SHP2 inhibition has direct effects on the TME that are important in antitumour immune responses. However, we still do not know if these additional effects in the myeloid cells are responsible for the enhanced infiltration of T cells, as SHP2i could also directly affect T cell activity. This complex interaction between the immune populations would require extensive and long-term investigation to address.

6. More detailed single cell RNA-Seq studies would no doubt have shed critical novel insights into sub-populations of myeloid and lymphoid cells that mediate the differential effects of RMC-4998, RMC-4550 and their combination and would further have allowed assessment of marker expression in individual cell populations rather than in bulk tumors.

We agree that single cell RNAseq studies would help to understand better the effects of G12C(ON) and/or SHP2 inhibitors in the different immune populations. However, these experiments were not possible in the timeline of these revisions. As indicated in the previous comment, we have deepened the characterization of the myeloid compartment by designing and performing multiparametric spectral flow cytometry using 36 markers. The new data is shown in Figure 5. These experiments led to the identification of SHP2-specific effects on macrophages described above. We also show effects induced specifically by the combination treatment on conventional dendritic cells (cDC1s), which have increased expression of MHCII molecule H-2Kb, indicative of activation and potential cross presentation to CD8 T cells (Fig. 5e).

7. Is the added effect specific to CTLA-4 blockade or is it observed also with inhibitors of other immune checkpoints such as TIM-3, LAG-3 or 4-1BB agonist antibodies?

That is a great suggestion, but we do think that this would be moving beyond the scope of a single paper. However, given that the combination of G12Ci + SHP2i increases inflammation and induces changes in several immune subpopulations we do believe there is potential to be combined with other types of immunotherapies, but due to limitations to the scale at which these complex in vivo experiments can be carried out in the time frame available, we have decided to focus only on those immune checkpoint blockade therapies that have been approved for the treatment of lung cancer.

8. *In the 3LL-ΔNRAS orthotopic model, the quadruple treatment arm with RMC-4998+RMC-4550+aPD-1+aCTLA-4 is missing. The argument that such a combination may not be tolerable in patients is plausible but based on the authors previous results with anti-CTLA-4 this is an important arm to include and would further have addressed the question as to whether the effect of CTLA-4 blockade is limited to the subcutaneous models.*

To address this question, we have now added the quadruple combination arm in Figure 7g-h. Although there is not a significant difference between the quadruple RMC-4998+RMC-4550+aPD-1+aCTLA-4 and triple RMC-4998+RMC-4550+aCTLA-4 treatments in terms of overall survival, the quadruple combination resulted in a higher percentage of tumours regressing in size by CT scan after 4 weeks of treatment and in the generation of two tumour-free mice. The mice showed some clear signs of toxicities with the quadruple combination that meant that obtaining adequately sized cohorts to achieve statistical significance was challenging, so we would rather not draw too strong conclusions from these data.

Reviewer #2

The manuscript entitled “Combining RASG12C(ON) inhibitor with SHP2 inhibition sensitises immune excluded lung tumours to immune checkpoint blockade” by Anastasiou et al. investigates novel drug combinations for refractory NSCLC. Traditional KrasG12C inhibitors such as adagrasib and sotorasib bind to the GDP bound KrasG12C and keep Kras in its inactive state. These are denoted as KrasG12C (OFF) inhibitors. Resistance to these inhibitors is common when used in the clinic to treat NSCLC and tumors recur within 7 months. Feedback reactivation of upstream RTKs, which result in wt or mutant Kras activation is a common resistance mechanism and so is the increased expression of mutant Kras. Thus, new inhibitors and or combinations of new therapies are needed to prevent resistance and treat refractory NSCLC.

In this study, the authors utilized a new KrasG12C inhibitor, which is selective for the active state of RASG12C(ON). Furthermore, the authors rationalize the combination with a SHP2 inhibitor, which should prevent feedback RTK signaling thereby preventing resistance. Lastly, the authors investigate combination of the two compounds with ICB in NSCLC mouse model of various immunogenicity. To my knowledge these combinations has not been tested and is highly relevant as these KrasG12C ON inhibitors are currently tested in clinical trials.

The manuscript is well written and logically constructed. The authors investigate single agents, combination therapy as well as sensitization to ICB. Two models are utilized, which have previously been shown to be immunogenic (KPARG12C) or immune excluded (3LL-ΔNRAS) to test their hypotheses. The authors conclude that relapse can be prevented with the combo in immunogenic tumors and induce an immune memory and that combo sensitizes to ICB in immune excluded ‘cold’ tumors by remodeling of the TME. The data is well presented and generally sound, however, since only two different models were used the question remains whether the therapeutic responses observed are dependent on differences in the TME of the two models or just cell line specific.

This is a very relevant concern. We have addressed this comment by replicating our observations in two additional mouse models of KRAS-mutant driven lung cancer.

To mimic an immunogenic model with reduced sensitivity to KRAS^{G12C} inhibition (G12Ci) we have generated a new cell line that was isolated from one KPAR^{G12C} orthotopic tumour that progressed on continued adagrasib treatment (details been added in the Methods section, the line is referred to as KPAR.M7). Using this less sensitive cell line we now observe an additional benefit from the addition of SHP2i to the combination of RMC-4998 + anti-PD-1, compared with the parental KPAR^{G12C} cells. In fact, we only generated durable complete regressions in the triple combination. This experiment has been added to the new Figures 3e, 3f and S3b. These results strengthen the importance of the triple combination for the treatment of those tumours with partial responses to KRAS^{G12C} inhibition and/or anti-PD-1 therapies.

We have also tested the combination in a syngeneic transplantable model (KPB6^{G12C}) derived from KP GEMM mice (KRAS^{G12D/-};Trp53^{-/-}). We have previously shown that, in contrast with the KPAR immunogenic model, KPB6 tumours are significantly infiltrated with neutrophils and show low infiltration of T cells (PMID: 35930804, reference #26 here). Although treatment with KRAS^{G12C} inhibitors increases T cell infiltration, this is not enough to sensitise these tumours to anti-PD-1 treatment (PMID: 3585784, reference #24 here). New Figure 4f shows that the addition of SHP2i to the combination sensitises tumours to anti-PD-1 and enhances tumour growth inhibition, which validates the benefit of the triple combination in a third model. However, the combination is not able to generate complete regressions in this model, probably due to low number of clonal tumour neoantigens.

It is unclear why expression of KrasG12C in presumably same lung epithelial cell origin results in an immunogenic or immune cold tumor. The authors show that the immune cold model is refractory to any ICB therapy and refer to previously published data about immunogenicity of the other model however detailed TME is not assessed by histology or flow cytometry prior to treatment in both models. It remains to be investigated why the murine KPARG12C is immunogenic and the 3LL-ΔNRAS is not. It is interesting that PD-1 therapy is not working in the immune excluded model as the authors show that PDL-1 expression appears in 40% of TAMs express PD-L1 in the vehicle treated? Is this because of the low number of T cells? A discussion of this would be helpful.

KPAR^{G12C} and 3LL Δ NRAS models have been extensively characterised in previous publications from our lab (PMID: 31534020 (ref 58), PMID: 35930804 (ref 26), PMID: 35857848 (ref 24), PMID: 37046094 (ref 48), PMID: 31898484) in terms of SNV content, immune infiltrate and response to KRAS inhibition and ICB response. Importantly, the baseline TME of these two models is completely different, with KPAR^{G12C} tumours being predominantly lymphocyte infiltrated in contrast to the 3LL- Δ NRAS model, which has a high infiltration of myeloid cells. We have now carried out immune profiling by FACS to compare both cell lines: this has now been included as new Supplementary Figure 5a. We have also added immunofluorescence analysis of CD8 T cells in tumours from both models in new Figure 3c and Supplementary Figure 6a. The differences in the TME could be partially explained by the origin of the cells. The KPAR^{G12C} model derives from a KP GEMM background (KRAS^{G12D/-};Trp53^{-/-}) that was also engineered to over-express APOBEC3B in order to increase tumour mutational burden (as detailed in PMID: 35930804 (ref 26). It was also derived from a Rag1^{-/-} background and therefore not exposed to T cell and B cell immune selection pressure to develop immune evasion mechanisms. It is re-expressing endogenous retroviral envelope protein, which we found is largely responsible for its immunogenicity and thus partial responsiveness to ICB (PMID: 35930804 (ref 26). Similar expression of endogenous retroviral envelope proteins has been documented in human lung adenocarcinoma patients (PMID: 37046094 (ref 48). We have extended the description of the KPAR^{G12C} model in the results section to try to explain this better. In contrast, KPB6 cells, which have been used in a new experiment (Figure 4f), derive from an immune competent KP background and present low T cell infiltration, do not respond to ICB, have very low tumour mutation burden and do not express retroviral elements (PMID: 37046094). Finally, the 3LL- Δ NRAS model derives from a spontaneously occurring lung adenocarcinoma from a C57/Bl6 mouse and has been passaged serially in syngeneic mice in a cell line that is highly immune evasive, refractory to lymphocyte infiltration and unresponsive to immunotherapy. The lack of T cell infiltrate may explain why even though there is a significant portion of TAMs expressing PD-L1, 3LL- Δ NRAS tumours are still resistant to anti-PD-1 therapy as pointed out by the reviewer. Overall, our previous research shows that there are many factors that contribute to response to therapies harnessing the immune system.

Nevertheless, the combination of the KrasG12C (ON) inhibitor with SHP2 inhibitor sensitizes to ICB in both models and appear to induce an adaptive immune response in both, thus the title and conclusion should be rephrased to include these findings.

The reviewer is right in this remark. We have modified the title, abstract and conclusion to indicate that the combination of G12C(ON)i and SHP2i can sensitise to ICB without specify the individual TME.

In general, there a several minor concerns pertaining to labeling and inclusion of statistical significance and the route of administration of all drugs/antibodies, which is omitted in Fig. legend, but found in methods.

Minor concerns:

- *Figure 1A: stats are missing, which makes it difficult to conclude that efficacy of one drug is better than the other.*

Statistics have been added.

- *Figure 1C: “combination of RMC-4550 with RMC-4998 prevented the rebound of ERK phosphorylation at later time points (24-48h)”. There appears to be pERK expression at 24-48hr in the CALU1 cells, so this statement should be toned down.*

This has now been modified to indicate that the reduction is not complete.

- *Figure 1D: there are no stats for viability graphs and thus it appears like an overinterpretation to state that the “combination of RMC-4550 with RMC-4998 prevented the rebound of ERK phosphorylation at later time points (24-48h) and resulted in a stronger reduction in cell viability”.*

Statistics have been added.

- *Figure 1E: stats are missing.*

Statistics have been added.

- *Figure 1H: first panel, hard to read since the first two lines indicating significance are at the same height. Also, there doesn't appear to be a statistical significance of the combination when compared to single agent treatment alone. Needs to be added to the text and should be indicated as ns in graph.*

Thank you for pointing this out. We have now adjusted the lines indicating significance to avoid confusion. While we agree that adding the non-significant comparisons to the graphs would be beneficial, the number of possible comparisons causes overlapping lines, making it difficult to distinguish between the groups being compared. For this reason, we have decided to add only the significant comparisons. One sentence indicating that “Only significant comparisons are shown” has been added to the legends containing multiple comparisons.

The reviewer is right that there is not statistical significance of the combination compared to single agent. We have clarified this in the text.

- *Figure legends 2-6, route of injection for drugs and ICB and dosing schedule for ICB are missing from the legend, but are listed in methods section.*

We have added the dosing schedule and treatment concentrations in all the legends. However, we have decided to describe the route of administration and formulation protocols in the methods section.

- *Why does tumor volume graph in Figure 2A go out 45 days but tumor volume graphs with indicated CR go out to 70 days in Figure 2B?*

Figure 2A shows the average of tumour volumes for each group. On day 45, some mice with relapsing tumours had to be culled because they reached the maximum volume allowed by the animal license or have ulcerated before reaching this endpoint. Therefore, averages could not be calculated after this point. Figure 2B represents the individual tumour volumes until the end of the experiment (day 70) to show that the complete responders remained tumour-free more than 30 days after treatment withdrawn. We have clarified this in the legend.

- *Figure 2C: stats missing. Looks like the combination displayed higher activity?*

The reviewer is right in this remark. The average of tumour volume and the statistical significance are shown in supplementary Figure 2B. This has now been indicated in the legend.

- *Figure 2D: stats for comparing single agent with combination are not indicated. Please add ns to the graph and mention in the result section.*

As indicated above and for consistency with all the figures, we have now added one sentence in the legend stating “Only significant comparisons are shown”.

- *Figure 2F: CR should be separated from the survival, way too confusing and can be inserted as a table. Also, neither in Figure legend nor in methods it is indicated whether appropriate IgG control was used in the groups not receiving aPD-1.*

To improve the visibility of the data we moved the number of complete responders from the graph legend to the graph itself. This figure is now labelled as Figure 3d.

We have also added the appropriate IgG controls used in the methods section.

- *Figure 3A: stats are described in the fig legend but not in panel.*

We have now stated in the legend that “Only significant comparisons are shown”.

- *Figure 3B: the lines and stars indicating significance are somewhat overlapping and hard to tell what groups are being compared.*

We have now modified the legend of the graph to improve the identification of the stats. This is now Figure 4b.

- *Figure 3B-E: The combo group with CTLA-4 alone should be included in the Figures 3B-E and stats should be added. Looking at the CTLA-4 data in Figs. Could it be that the effect is CTLA-4 dependent and not additive as implied by authors. Figures 3C-E can be moved in supplemental and instead show table with CR and stats. Instead of Figure 3C-E, the Figure S3c,d data could be moved into the main body.* The experiment shown in Figure 3b-c (Now Figure 4) did not include the combo group with CTLA-4. We have now added one new experiment including this group plus combo+PD-1 and combo+PD-1+CTLA-4 as comparison (Supplementary Figure 4d). Following the reviewer's suggestion we have moved to supplementary the combinations of ICB with RMC-4998 or RMC-4550 and we have created a new figure with the percentage of complete responders for each treatment (Figure 4e). As pointed by the reviewer, this figure indicates that the combination with ICB with single treatment of G12C(ON)i or SHP2i is CTLA-4 dependent. However, this effect is additive when tumours are treated with the combination of targeted therapies.

- *Figure 4C statistical significance comparisons should be the same as in Figure 4B, panel two: single agent alone compared to combo. Statistical sig. compared to vehicle could be indicated in Fig. legends.* This experiment (now Figure 5b) has been substituted by a new experiment using a larger panel of FACS antibodies for a deeper characterisation of the myeloid populations. Statistical significance for the new figures has been added.

- *Figure 5A, B: comparison of single with combo should always be indicated even if ns.* As indicated above, to avoid confusion between the lines showing significance and to maintain the consistency with all the figures, we have now added one sentence in the legend indicating that "Only significant comparisons are shown".

- *Figure 5A,B should indicate if data is from orthotopic tumors.* This has now been indicated in the legend.

- *Figure 6F: text line 352: "elevated infiltration and persistent proliferation of CD8+ and CD4+T cells (Fig. 6F). There is no statistical diff. between combo and combo plus ICB in CD8 T cells", please tone down the conclusion.* The conclusion has been toned down.

- *Was combination of PD1 and CTLA-4 tried in subcue model (Figure 6H)* To address this question, we have now added the quadruple combination arm in Figure 7g-h (previous Figure 6h-i). Although there is not a significant difference between both treatments, the quadruple combination resulted in higher percentage of tumours regressing after 4 weeks of treatment and in the generation of two tumour-free mice.

- *Figure 4 doesn't explain the CR observed in the different tx groups (in Figure 3) since there are very few differences between single agent and combo. Plz discuss appropriately.* This is a relevant concern. Lung and subcutaneous tumours have different tumour microenvironments and we and others have shown that this can result in different responses to ICB treatment (PMID: 36736322, PMID: 35930804 (ref 26)). Another reason for the differences in response is the difficulty in generating tumour-free mice in this aggressive cancer model grown orthotopically, as lungs contain multiple tumours with varying numbers between mice (with an average of 4.5 tumours per mouse). This is why we also include the tumour volume change after 4 weeks of treatment, as this measure is more representative of individual tumour changes, similar to the subcutaneous experiments. This has been clarified in the results section (page 13).

• Add to discussion that the observed different outcome between the groups in tumor volume and survival (Figure 6) could also be due to the difficulty of measuring tumor volume with microCT? This could be a factor, although our lab does have extensive experience analysing lung tumour volume changes using microCT scan and has published a protocol article detailing our methods (PMID: 36494493, ref 57). Due to the aggressiveness of the 3LL Δ NRAS model, mice are scanned every week, allowing for an exhaustive analysis of tumour volume changes during treatment. We believe that the cause of the different outcome between the changes in tumour volume and the survival may be due to the presence of multiple tumours per lung and the variable response of these tumours to the treatment. This has been clarified in the results section (page 13).

By way of an example, we show below for the reviewer the tumour volume (mm³) analysis of one mouse treated with the quadruple combination that showed different tumour responses.

scan 1	scan 2	scan 3	scan 4	scan 5	scan 6	scan 7	% change scan 4 vs scan 1
15.35	4.13	2.22	1.41	0.93	0.74	1.11	-93.94
2.63	0.74	0.65	0.61	0.62	0.72	0.82	-76.43
0.71	0.07	0.09	0.06	0.06	undetectable due to collapsed lobe		-91.55
0.71	0.24	0.57	0.71	0.56	1.28	4.34	-21.13
0.55	0.13	0.09	0.01	0	0	0	-100.00
0.51	0.28	0.24	0.17	0.21	0.4	1.50	-58.82
0.49	0.16	0.13	0.13	0.16	0.16	0.43	-67.35
2.94	0.54	0.67	1.17	3.65	8.02	22.81	24.15
0.87	0.13	0.11	0.08	1.46	3.63	6.82	67.82
0.81	0.38	0.42	0.6	1.23	4.51	15.42	51.85

Table for reviewer. Tumour volume (mm³) of individual 3LL Δ NRAS tumours in one mouse treated with the quadruple combination (KRAS G12Ci+SHP2i+anti-PD-1+anti-CTLA-4. Each row represents one tumour. Scan 1 indicates the tumour volume at the start of the treatment, with the mouse scanned once a week. The last column shows the change in tumour volume between initial tumour volume (scan 1) and the scan after 4 weeks of treatment (scan 4) as shown in Figure 7h.

Reviewer #3

This is an important characterization study of a new G12C inhibitor in combination with a Shp2 inhibitor as new G12C inhibitors are needed in the clinic. Study is highly rigorous. Some aspects of the study need strengthening. There are two novel drugs, many different combinations resulting in different immune phenotypes. Differences in the response of subcutaneous vs orthotopic models to the treatments are not discussed or analyzed.

One question relates to the model used in most of the experiments-KPAR-G12C, this model should be described.

The KPAR^{G12C} model derives from a KP GEMM background (KRAS^{G12D/-};Trp53^{-/-}) that was also engineered to over-express APOBEC3B in order to increase tumour mutational burden, as detailed in a previous paper from our lab (PMID: 35930804 (ref 26)). It was also derived from a Rag1^{-/-} background and therefore not exposed to T cell and B cell immune selection pressure to develop immune evasion mechanisms. It is re-expressing endogenous retroviral envelope protein, which we found is largely responsible for its immunogenicity and thus partial responsiveness to ICB (PMID: 35930804 (ref 26)). Similar expression of endogenous retroviral envelope proteins has been documented in human lung adenocarcinoma patients (PMID: 37046094 (ref 48)). We have extended the description of the KPAR^{G12C} model in the results section to try to explain this better. We have also added a FACS immune profiling comparing the KPAR^{G12C} and 3LL ΔNRAS tumours at basal level (Supplementary Figure 5a) and immunofluorescence analysis of CD8 T cells for both models in new Figure 3a and Supplementary Figure 6a.

Subcutaneous tumors are relatively small when treatments are initiated.

We select the size of the 3LL-ΔNRAS subcutaneous tumours at the start of the treatment to be relatively small (average of 100 mm³) compared to other cell lines due to the aggressiveness of the tumours formed by this cell line growing in vivo. As can be seen in the Vehicle treated tumours, these can double in size within 48 hrs. For the rest of the cell lines used in the subcutaneous tumour models, the growth was somewhat less aggressive, so the average tumour volume at the start of the treatment was chosen to be above 150 mm³.

Volume of orthotopic tumors are not shown.

The variability of lung tumours renders the results difficult to interpret when raw orthotopic volume is shown, which is why we normalise to the starting volume to get the % of volume change. Lungs contain multiple tumours with varying numbers and size between mice (with an average of 4.5 tumours per mouse). Mice are scanned every week and are randomised based on the initial tumour burden. This has now been clarified in the methods section.

To illustrate the complexity of the tumour growth and response in this model, in the table below for the reviewer we show the tumour volume (mm³) analysis of one mouse with high tumour burden treated with the quadruple combination that showed different tumour responses.

scan 1	scan 2	scan 3	scan 4	scan 5	scan 6	scan 7	% change scan 4 vs scan 1
15.35	4.13	2.22	1.41	0.93	0.74	1.11	-93.94
2.63	0.74	0.65	0.61	0.62	0.72	0.82	-76.43
0.71	0.07	0.09	0.06	0.06	undetectable due to collapsed lobe		-91.55
0.71	0.24	0.57	0.71	0.56	1.28	4.34	-21.13
0.55	0.13	0.09	0.01	0	0	0	-100.00
0.51	0.28	0.24	0.17	0.21	0.4	1.50	-58.82
0.49	0.16	0.13	0.13	0.16	0.16	0.43	-67.35
2.94	0.54	0.67	1.17	3.65	8.02	22.81	24.15
0.87	0.13	0.11	0.08	1.46	3.63	6.82	67.82
0.81	0.38	0.42	0.6	1.23	4.51	15.42	51.85

Table for reviewer. Tumour volume (mm³) of individual 3LL ΔNRAS tumours in one mouse treated with the quadruple combination (KRAS G12Ci+SHP2i+anti-PD-1+anti-CTLA-4). Each row represents one tumour. Scan 1 indicates the tumour volume at the start of the treatment, with the mouse

scanned once a week. The last column shows the change in tumour volume between initial tumour volume (scan 1) and the scan after 4 weeks of treatment (scan 4) as shown in Figure 7h.

Does RMC-G12C inhibitor delays resistance as compared to sotorasib in vivo in KPARC model?

We examined the effect of the approved KRAS^{G12C}(OFF) inhibitor adagrasib and compared with RMC-4998 using the 3LL-ΔNRAS subcutaneous model and did not observe any major differences in response (new Supplementary Figure 4e). Head-to-head comparisons of two drugs with closely related but not identical mechanisms of action are complicated by a wide range of factors and we would be very cautious about drawing too much by way of conclusion from these limited data. However, in these experiments, adagrasib is not inferior to RMC-4998, nor is adagrasib plus RMC-4550 inferior to RMC-4998 plus RMC-4550. This might imply that the cancer-cell independent effects of SHP2 inhibition acting on cells in the TME, especially macrophages, as first described in reference #30 (PMID: 32350067) and further elucidated in this manuscript, may be more important than the cancer cell autonomous effects of SHP2 inhibition in deepening the amplitude and duration of inhibition of KRAS signaling by KRAS G12C (OFF) inhibitor. Moreover, we now show that the combination of KRAS^{G12C} inhibitor (G12Ci) and SHP2 inhibition also sensitizes tumours to anti-PD-1 and generates durable complete responses when the G12C(OFF) inhibitor adagrasib is used (new Supplementary Figure 4f), similarly to the G12C(ON) inhibitor. This is highly relevant as it suggests that this combination could also be used with the multiple KRAS^{G12C}(OFF) inhibitors currently in clinical trials, some of them already testing the dual combinations G12Ci+SHP2i and G12Ci+anti-PD-1. This has now been indicated in the discussion.

Would this drug in combination with the Shp2 similarly induce anti-tumor immunity in a non-G12C driven model?

Although we have not tried the combination of SHP2 inhibitor RMC-4550 with compounds targeting other mutant KRAS isoforms, we have immuno-profiled KRAS^{G12D/-};TP53^{flox/flox} lung cancer GEMM model after RMC-4550 treatment for 7 days. We observe a similar infiltration of lymphocytes (see figure below) as seen in other models compared with Vehicle which would suggest that antitumour immunity could be induced by combination with others KRAS-mutant selective inhibitors. We have added a sentence in the discussion suggesting that this could be an interesting combination to test. However, we feel that these experiments go beyond the current scope of this paper, so would prefer not to include these data.

Figure for reviewer. FACS immune profiling of KRAS^{G12D/-};TP53^{flox/flox} lung tumours treated with 30mg/kg RMC-4550 for 7 days.

Mechanistic insights into the reasons for immune cell sculpting in both KPARC and 3LL needs strengthening. Is there a direct connection and if so, what is the connection?

In order to strengthen the mechanistic understanding of the events studied here, we have added multiplex immunofluorescence analysis of CD8 T cells for both models in new Figure 3a and Supplementary Figure 6a. In addition, due to the high infiltration of myeloid cells in the 3LL ΔNRAS model we decided to characterise better the effects of SHP2i and G12C(ON)i on the myeloid populations by designing a panel of 36 markers for multiparametric spectral flow

cytometry (details in Methods and new Supplementary Figure 9). New data is shown in Figure 5. We show that SHP2i treatment leads to a greater proportion of CD86⁺ MHCII⁺ and PD-L1⁺ interstitial macrophages compared to G12C(ON)i, suggesting that some effects driven by SHP2 inhibition are independent of the cancer-cell intrinsic KRAS signalling. By incorporating more markers associated with myeloid function we show that SHP2i, but not G12C(ON)i, induces maturation and antigen presentation. These enhanced effects of SHP2i in the interstitial macrophage population are correlated with the increase in T cell infiltration (Fig. 6a), again reinforcing the suggestion that SHP2 inhibition has direct effects on the TME that are important for anti-tumour immune responses. We also show effects induced specifically by the combination treatment on conventional dendritic cells (cDC1s), which have increased expression of MHC-I molecule H-2Kb, indicative of activation and potential cross presentation to CD8 T cells (Fig. 5e).

Would the 4-drug combination cause enhanced toxicity in patients since Kras inhibitors in combination with pembroluzimab causes toxicities. Along these lines, in the orthotopic models in figures 1 and 6 (Kpar vs 3LL): there is not an obvious difference between RMC4998 vs RM4998+RMC4550 groups. We agree with the reviewer about the potential toxicity of the quadruple combination. We discuss the potential toxicities of these combinations at the end of the discussion. First generation KRAS G12C inhibitors have shown significant toxicity in combination with anti-PD-1 ICB therapy, but this does appear to be off target for KRAS, so there is hope that subsequent more specific inhibitors of KRAS may combine better with ICB. However, we must accept that four-way drug combinations of this sort are unlikely to be a practical prospect in the clinic in the immediate future.

We also agree that in Figure 3a (previous Figure 2a) KPAR^{G12C} tumours do not show a significant difference in tumour regression after one week of treatment. However, as we show in Figure 3b, differences are observed at longer time points, which result in an increased survival (Figure 3d), although most of the survival benefit could be achieved with the combination of RMC-4998 plus anti-PD-1. We hypothesise that the lack of difference between RMC-4998 vs RMC-4998+RMC-4550 when anti-PD1 was added was due to the strong sensitivity of the KPAR^{G12C} tumours to RMC-4998. To test this, we have developed a new cell line with partial response to G12Ci. This cell line has been isolated from one KPAR^{G12C} orthotopic tumour that progressed on continued adagrasib treatment (details have been added in the Methods section, the line is termed KPAR.M7). Using this cell line, we now observe an additional benefit with the different combinations, with the triple combination (KRAS G12C(ON)i+SHP2i+anti-PD-1) being the only treatment that generated durable complete regressions in this model. This experiment has been added to the new Figures 3e, 3f and S3b.

In the case of the 3LL Δ NRAS tumours, because they are less sensitive to RMC-4998, we observe significant tumour regression with the combination (Figure 7a) and increased survival (Figure 7b). However, only the combination of RMC-4998 plus RMC-4550 is able to sensitise tumours to ICB, demonstrating the importance of this combination in this immune evasive lung cancer model.

What is the antigen in non-immunogenic -3LL tumors? If there are antigens, then does the baseline tumor microenvironment dictate the differences observed between KPARC and 3LL with the treatments?

The 3LL Δ NRAS has been extensively characterised in previous publications from our lab (PMID: 35857848 (ref 24), PMID: 34625563, PMID: 31898484) in terms of SNV content, immune infiltrate and response to KRAS inhibition and ICB response. Specifically, 3LL- Δ NRAS carry many in silico predicted neoantigens (~2000) but have epigenetically suppressed H-2Kb to evade presentation of these neoantigens in an MHCI manner (PMID: 35857848 (ref 24)). Four strong neoantigens have been experimentally verified in this model (PMID: 32115148). As pointed out by the reviewer, the existence of these neoantigens indicates that there is potential

for inducing an immune response towards this model, however at baseline there are no lymphocytes present in the tumour area, with them rather being excluded to the tumour border, indicative of immune evasion mechanisms acting in the TME. We have added an immunofluorescence analysis of CD8⁺ T cells that illustrates this (Supplementary Figure 6). In contrast, the KPAR^{G12C} model was raised in a Rag1^{-/-} background and therefore not being exposed to T cell and B cell immune selection pressure to develop immune evasion mechanisms. As indicated above, we have previously shown that KPAR tumours express retroviral elements rendering them visible to the immune system and thus partially respond to ICB (PMID: 37046094). All these factors determine the baseline TME composition and consequently differences with treatment that harness different components of the immune system. To show the differences in the TME composition, FACS immune profiling comparing both cell lines has been included as Supplementary Figure 5a.

Authors mention changes in B and NK cells with the combination treatment. What is the role of B cells in this therapeutic response and do they contribute to the response mediated by Kras G12C and Shp2 inhibitors?

This is a very interesting question. To explore this further, we tried depleting B cells with an anti-CD20 depleting antibody. However, we did not see any differences in the survival in mice treated with RMC-4998 + RMC-4550. After immunophenotyping the different treatment groups, we observed that even though treatment with anti-CD20 antibody was able to deplete B cells efficiently, it was unable to deplete antibody-secreting plasma cells. In addition, the length of the anti-CD20 treatment (6 days) was likely insufficient to reduce circulating anti-tumour cell antibody levels. Therefore, we cannot exclude a role for the humoral arm of the immune system in the response to G12C(ON) or SHP2 inhibitors. Due to the inconclusive nature of these experiments, we decided not to include these data in the manuscript.

Some of the individual treatment arm tumor measurement data can be moved to supplementary figures and immune characterization of KPARC model should be moved to main figures.

Following the reviewer's suggestion we have moved to supplementary the combinations of ICB with RMC-4998 or RMC-4550 in old Figure 3, now in Supplementary Figure 4b. Moreover, we have created a new figure with the percentage of complete responders for each treatment (Figure 4e). We have also added a new experiment of immune characterization of KPAR^{G12C} tumours showing infiltration and activation of CD8⁺ T cells by multiplex immune fluorescence (new Figure 3c).

For the immune characterization some more detail is needed,

As indicated in the previous comments, we have added additional immune characterization for both models. These include FACS immune profiling comparing the basal TME of KPAR^{G12C} and 3LL Δ NRAS tumours (Supplementary Figure 5a), immunofluorescence analysis of CD8 T cells for both models (Figure 3a and Supplementary Figure 6a) and a new multiparametric spectral flow cytometry panel analysing 36 markers for deeper characterization of the changes in myeloid populations after treatment with RMC-4998 and/or RMC-4550 in the 3LL Δ NRAS orthotopic model (Figure 5).

are the different groups normalized by the tumor amount?

The immune characterisation of tumours using flow cytometry is done on isolated lung tumours and not on whole lung. However, due to the small size of the tumours after treatment, we cannot immune profile them individually. Therefore, as indicated in the Methods section, all tumours in one lung were pooled together for immune profiling.

REVIEWERS' COMMENTS

Reviewer #1 (Remarks to the Author):

In response to the reviewer's comments the authors have conducted extensive additional analyses which have considerably strengthened the manuscript. I have no further critiques.

Reviewer #2 (Remarks to the Author):

The authors have appropriately addressed reviewer's concerns and have included new data addressing previous concerns. Furthermore, the authors added experimental outlines and statistics where requested. This revised manuscript overall is strengthened by the additional data and appropriate for publication.

Reviewer #3 (Remarks to the Author):

Authors addressed most of the comments. Authors indicate that quadruple combination in revised figure 7 caused toxicity in mice in the response letter. They should include the details of these toxicities at least in the text. It is not common for mice to show IRAE.

Authors had shown that cold tumor LLC could be sensitized to ICB with the two RMC drugs. In light of new figure 4f with the new model, authors need to change their statement including in the abstract since this model does not show CR.

It is not mentioned in the text that the immunogenic G12C line is expressing Apobec3b. While authors cite their previous manuscript, they should write here too.

Response to Reviewers' Comments

Reviewer #1

In response to the reviewer's comments the authors have conducted extensive additional analyses which have considerably strengthened the manuscript. I have no further critiques.

Reviewer #2

The authors have appropriately addressed reviewer's concerns and have included new data addressing previous concerns. Furthermore, the authors added experimental outlines and statistics where requested. This revised manuscript overall is strengthened by the additional data and appropriate for publication.

Reviewer #3 (Remarks to the Author)

Authors addressed most of the comments. Authors indicate that quadruple combination in revised figure 7 caused toxicity in mice in the response letter. They should include the details of these toxicities at least in the text. It is not common for mice to show IRAE.

Although we did not see evident toxicity with the quadruple combination, we did see toxicities in 2/12 mice in the triple combination (KRAS inhibitor, SHP2 inhibitor, anti-CTLA-4). This is now stated in the results text.

Authors had shown that cold tumor LLC could be sensitized to ICB with the two RMC drugs. In light of new figure 4f with the new model, authors need to change their statement including in the abstract since this model does not show CR.

It is correct that the immune desert model, KPB6, does not show complete responses, whereas the immune excluded model, 3LL, does. The cold/hot designation, in which both these models are considered cold, may be somewhat confusing, so we have changed the abstract text to remove that description and clarify this point.

It is not mentioned in the text that the immunogenic G12C line is expressing Apobec3b. While authors cite their previous manuscript, they should write here too.

Although the KPAR cell line derived from a tumour that was engineered to over-express APOBEC3B, the established cell line has lost APOBEC3B expression, as is shown in Fig. 3A of PMID: 35930804. However, to clarify, we have specified in the results section that KPAR^{G12C} cell line derives from a KRAS^{LSL-G12D/-}Trp53^{f/f} Rosa26^{hA3Bi} Rag1^{-/-} mouse background.